# Causal Sufficiency and Necessity Improves Chain-of-Thought Reasoning

**Xiangning Yu**[1,6][*] **Zhuohan Wang**[2][*] **Linyi Yang**[3], **Haoxuan Li**[4],
**Anjie Liu**[5], **Xiao Xue**[1][†] **Jun Wang**[3], **Mengyue Yang**[6][†‡]

[1]Tianjin University     [2]City University of Hong Kong
[3]University College London     [4]Peking University
[5]The Hong Kong University of Science and Technology (Guangzhou)     [6]University of Bristol

## Abstract

Chain-of-Thought (CoT) prompting plays an indispensable role in endowing large language models (LLMs) with complex reasoning capabilities. However, CoT currently faces two fundamental challenges: (1) Sufficiency, which ensures that the generated intermediate inference steps comprehensively cover and substantiate the final conclusion; and (2) Necessity, which identifies the inference steps that are truly indispensable for the soundness of the resulting answer. We propose a causal framework that characterizes CoT reasoning through the dual lenses of sufficiency and necessity. Incorporating causal Probability of Sufficiency and Necessity allows us not only to determine which steps are logically sufficient or necessary to the prediction outcome, but also to quantify their actual influence on the final reasoning outcome under different intervention scenarios, thereby enabling the automated addition of missing steps and the pruning of redundant ones. Extensive experimental results on various mathematical and commonsense reasoning benchmarks confirm substantial improvements in reasoning efficiency and reduced token usage without sacrificing accuracy. Our work provides a promising direction for improving LLM reasoning performance and cost-effectiveness. The code is available at: `https://github.com/yxn9191/causalmath`.

## 1 Introduction

Large Language Models (LLMs) have demonstrated impressive advancements in complex reasoning tasks, significantly attributed to the adoption of Chain-of-Thought (CoT). CoT prompting guides models to generate intermediate reasoning steps, thereby enhancing performance in areas such as arithmetic problem-solving and commonsense reasoning [43, 17, 8]. Despite these improvements, CoT reasoning faces two fundamental challenges: **(i) Sufficiency**: ensuring that the generated intermediate steps comprehensively support the conclusion [50, 3, 33], and **(ii) Necessity**: identifying which steps are indispensable for the soundness of the final answer [7, 54]. Figure 1a illustrates three common reasoning patterns frequently observed in LLMs, exemplified here using a GSM-8k [10] question: (1) *Sufficient but Unnecessary*, where redundant steps reduce reasoning efficiency; (2) *Necessary but Insufficient*, in which incomplete reasoning fails to reach the correct answer; and (3) *Sufficient and Necessary*, the ideal case that balances correctness and conciseness. These examples highlight the impact of reasoning inefficiencies—especially "overthinking", where unnecessary steps may hinder rather than help model performance.

---

[*]Equal contribution.
[†]Corresponding authors.
[‡]Project leader

39th Conference on Neural Information Processing Systems (NeurIPS 2025).

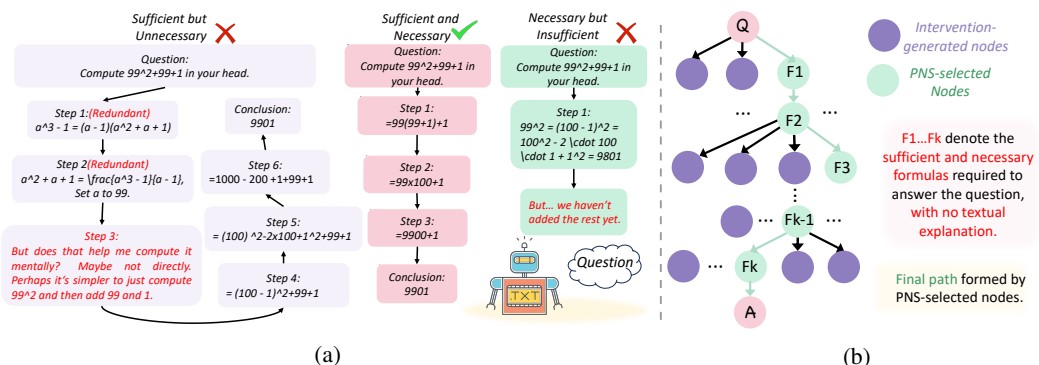

Figure 1: (a) Illustration of three reasoning types—*Sufficient but Unnecessary*, *Necessary but Insufficient*, and *Sufficient and Necessary*—based on actual model-generated responses to a GSM-8k question: "Compute $99^2 + 99 + 1$ in your head." (b) Path selection process using our method. Purple nodes denote CoT steps obtained through causal intervention (rollout), while green nodes indicate the minimal steps satisfying both sufficiency and necessity.

Recent research on Chain-of-Thought (CoT) reasoning has addressed **Sufficiency** by introducing strategies such as self-consistency decoding [59] and iterative refinement methods like Self-Refine [40], aiming to ensure intermediate steps comprehensively support final answers [27, 48, 17]. Concurrently, efforts targeting **Necessity** have developed pruning techniques, such as addressing the "overthinking" by reducing the token length [7, 36]. Chain-of-Draft prompting [67] and identify critical reasoning steps [13], to reduce redundancy in reasoning paths [42, 62, 52, 45]. However, none have utilized rigorous mathematical analyses based on sufficient and necessary conditions [46] to evaluate and prune reasoning paths. These methods predominantly rely on correlation-based metrics (e.g., attention weights, likelihood scores, or ablation accuracy), which may misleadingly associate frequent or prominent steps with correctness without verifying true causal impact [4]. Consequently, correlation alone is insufficient for reliably distinguishing genuinely necessary or sufficient reasoning steps, highlighting the need for causal frameworks to rigorously assess their logical contributions.

To jointly address the sufficiency and necessity of reasoning steps while ensuring logical and causal soundness, we introduce the concept of causal Probability of Necessity and Sufficiency (PNS) and redefine it for CoT reasoning framework. We theoretically analyse the identifiability of PNS in CoT. Based on the identifiability results, we develop a PNS-based evaluation algorithm to systematically reconstruct reasoning sequences by causal intervention (rollout) (shown in Figure 1b). Using this algorithm, we effectively reconstruct CoT responses from training data that explicitly meet causal sufficiency and necessity criteria, thus eliminating redundant steps without compromising—and potentially enhancing—answer accuracy. The reconstructed reasoning CoT then serve as causally-informed demonstrations, enabling LLMs to acquire causal reasoning capabilities via in-context learning and fine-tuning to improve the efficiency without sacrificing the accuracy. Empirical evaluations on mathematical reasoning benchmarks—including GSM-8k [10], MATH-500 [25], and AIME [44], as well as the CommonsenseQA [53] dataset—confirm that our approach significantly reduces reasoning redundancy while maintaining or improving prediction accuracy.

Our main contributions are as follows:

1. We propose a conceptual integration of Probability of Necessary and Sufficient causation (PNS) into CoT reasoning.

2. We introduce a novel bi-level optimization framework based on PNS for systematically constructing efficient and accurate CoT reasoning sequences.

3. We empirically validate our approach across diverse reasoning tasks, demonstrating substantial improvements in both efficiency and accuracy through optimized CoT traces used for in-context learning and supervised fine-tuning.

## 2   Related Work

**Reasoning Sufficiency Enhancement via CoT Optimization.** CoT reasoning [61] has significantly improved LLM performance on complex tasks, inspiring variants such as Tree-of-Thought [76], Graph-of-Thought [6], and DOTS [78]. Further developments include multimodal extensions [65], latent variable formulations [64], and dynamic memory usage [73]. Others enhance reasoning via self-correction [47], counterfactual fine-tuning [26], or prompt design [18]. Despite strong performance, many methods suffer from unnecessary verbosity, inefficient computation, or overthinking [62, 7]. Beyond textual reasoning, task-grounded reasoning frameworks such as ChessGPT [20] bridge policy learning and language modeling, revealing that reasoning sufficiency can also be optimized in structured decision domains.

**Reasoning CoT Redundancy.** Recent work targets CoT redundancy by compressing reasoning traces (e.g., C3oT [32], CoT-Valve [39], CCoT [9]), pruning superfluous steps [36], or using token-budget-aware reasoning [22]. Training-free approaches such as Kimi [54] and external thought injection [35] further optimize reasoning cost. SPIRIT [13] leverages perplexity to identify key reasoning steps, balancing accuracy and efficiency in both few-shot and fine-tuned CoT settings, while also generalizing well across models. Ton et al. [57] use conditional mutual information to quantify each step's contribution to the final answer, revealing failure patterns without requiring intermediate supervision. However, existing methods often prioritize brevity or representational efficiency, without explicitly enforcing causal sufficiency or necessity.

**PNS Theory in CoT Reasoning.** Extending Pearl [46], we introduce the Probability of Necessary and Sufficient causes (PNS) framework to CoT reasoning, applying PN and PS at the step level rather than the model level as in Hüyük et al. [26]. This enables causal pruning of redundant steps, yielding minimal yet faithful CoTs. Relatedly, Yang et al. [72] formalize invariant learning through PNS estimation, offering a principled view of causal sufficiency and necessity in representation learning. Building on this foundation, recent work on LLM causal reasoning [29] extends similar principles to in-context learning. Our approach generalizes these insights to step-level reasoning dynamics, providing a model-agnostic and theoretically grounded alternative to heuristic compression methods.

**Causal Necessity and Sufficiency in XAI.** Prior studies have leveraged causal necessity and sufficiency to explain model behavior. LENS [60] identifies necessary and sufficient output conditions; Darwiche and Hirth [14] compute sufficient reasons via Decision-DNNF circuits; Mothilal et al. [41] generate diverse counterfactuals; Beckers [5] formalize sufficiency-based explanations for fairness; and Galhotra et al. [21] propose LEWIS, a probabilistic counterfactual method. Recent progress in causal representation learning [71, 70] further explores disentanglement and invariance through structural causal models and counterfactual reasoning, demonstrating how causal principles enable robust and interpretable representations. Building on these ideas, our PNS evaluation extends necessity and sufficiency analysis to LLM reasoning chains, using counterfactual rollouts to assess the causal faithfulness of CoT traces and mitigate overthinking.

## 3   Defining Causal Necessary and Sufficiency in CoT

### 3.1   Chain-of-Thought (CoT) Reasoning

**Definition 1 (Chain-of-Thought (CoT) Reasoning [61])** *Given an input* $\mathbf{Q} = \mathbf{q}$, *the Chain-of-Thought (CoT) reasoning process generates the final answer* $\mathbf{A} = \mathbf{a}$ *by sequentially deriving intermediate reasoning steps* $\mathbf{S} = \{\mathbf{s}_1, \mathbf{s}_2, \ldots, \mathbf{s}_n\}$. *The probability of generating the answer given the question is defined as:*

$$P(\mathbf{A} = \mathbf{a} \mid \mathbf{Q} = \mathbf{q}) \propto \int \underbrace{P(\mathbf{a} \mid \mathbf{s}_1, \ldots, \mathbf{s}_n, \mathbf{q})}_{\text{Answer Generation}} \times \underbrace{\prod_{i=1}^{n} P(\mathbf{s}_i \mid \mathbf{s}_{<i}, \mathbf{q})}_{\text{CoT Generation}} \, d\mathbf{S}. \tag{1}$$

**Explanation:**   $P(\mathbf{A} = \mathbf{a} \mid \mathbf{Q} = \mathbf{q})$ is the final answer probability. $P(\mathbf{a} \mid \mathbf{s}_1, \ldots, \mathbf{s}_n, \mathbf{q})$ corresponds to the conditional probability of generating the final answer from the full reasoning trace. $\prod_{i=1}^{n} P(\mathbf{s}_i \mid \mathbf{s}_{<i}, \mathbf{q})$ models the sequential reasoning process. The integral marginalizes over all possible reasoning traces $\mathbf{S}$.

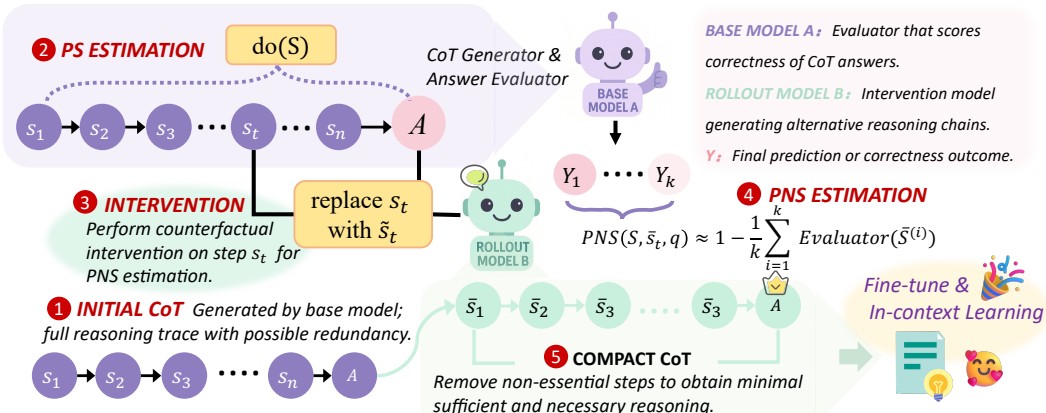

Figure 2: Causal Optimization Framework for CoT Reasoning. Our method identifies and retains only causally essential reasoning steps to form a compact CoT. (1) A base model generates the initial CoT trace, possibly containing redundant steps. (2) Sufficiency is estimated by checking if the full CoT leads to a correct answer. (3) For each step $s_t$, necessity is evaluated via counterfactual substitution $\tilde{s}_t$ using a rollout model, followed by answer scoring from the base model. (4) The Probability of Necessity and Sufficiency (PNS) is computed to measure causal contribution. (5) Non-essential steps are pruned to obtain a compact CoT, which is then used for fine-tuning or in-context learning.

## 3.2 Causal Necessary and Sufficiency in CoT

To rigorously characterize the causal significance of individual reasoning steps in Chain-of-Thought (CoT) reasoning, we propose formal definitions of causal sufficiency and causal necessity tailored to the structure and properties of CoT.

**Definition 2 (Sufficiency, PS)** *Sufficiency measures whether the reasoning chain* $\mathbf{S} = (\mathbf{s}_1, \ldots, \mathbf{s}_n)$ *is sufficient to produce the correct answer* $\mathbf{y}$. *Following the counterfactual definition in Pearl [46], the probability of sufficiency is defined as:*

$$\mathrm{PS}(\mathbf{S}, \mathbf{q}) = P\left(\mathbf{A}_{\mathrm{do}(\mathbf{S})} = \mathbf{y} \mid \mathbf{A} \neq \mathbf{y}, \bar{\mathbf{S}}, \mathbf{q}\right), \tag{2}$$

where $\mathrm{do}(\mathbf{S})$ means the intervention which set the value of chain variable as $\mathbf{S}$, $\mathbf{A}_{\mathrm{do}(\mathbf{S})}$ denotes the counterfactual answer had the reasoning chain $\mathbf{S}$ been used, and $\mathbf{A}_{\mathrm{do}(\bar{\mathbf{S}})}$ denotes the actual answer under the original reasoning $\bar{\mathbf{S}}$ (which could be null or incorrect). This captures the likelihood that inserting $\mathbf{S}$ would have changed an incorrect answer to a correct one.

**Definition 3 (Necessity, PN)** *Necessity quantifies whether a specific reasoning step* $\mathbf{s_t}$ *is required for producing the correct answer* $\mathbf{a} = \mathbf{y}$. *Inspired by the counterfactual definition of necessity [46], we define the probability of necessity as:*

$$\mathrm{PN}(\mathbf{S}, \bar{\mathbf{s}}_t, \mathbf{q}) = P\left(\mathbf{A}_{\mathrm{do}(\mathbf{s}_{<t}, \bar{\mathbf{s}}_t, \mathbf{s}'_{>t})} \neq \mathbf{y} \mid \mathbf{A} = \mathbf{y}, \mathbf{S}, \mathbf{q}\right), \tag{3}$$

where $\mathbf{s}_{<t}$ denotes the set of all correct reasoning steps before position $t$, $\bar{\mathbf{s}}_t$ represents an incorrect or corrupted variant of the original step $\mathbf{s}_t$. The counterfactual outcome $\mathbf{A}_{\mathrm{do}(\bar{\mathbf{s}}_t)}$ is defined as the model's predicted answer when $\mathbf{s}_t$ is replaced by $\bar{\mathbf{s}}_t$, and the subsequent steps $\mathbf{s}'_{>t}$ are generated conditioned on this modified reasoning trajectory.

**Definition 4 (Probability of Necessary and Sufficient Cause (PNS) in CoT)** *Given a reasoning chain* $\mathbf{S} = (\mathbf{s}_1, \ldots, \mathbf{s}_n)$ *that produces the correct answer* $\mathbf{A} = \mathbf{y}$, *and an alternative reasoning step* $\bar{\mathbf{s}}_t$ *at position* $t$, *let the counterfactual chain be defined as:*

$$\mathbf{S}' = (\mathbf{s}_{<t}, \bar{\mathbf{s}}_t, \mathbf{s}'_{>t}),$$

*where* $\mathbf{s}_{<t}$ *denotes the preceding steps, and* $\mathbf{s}'_{>t}$ *are subsequent steps possibly adapted to* $\bar{\mathbf{s}}_t$.

*Inspired by the counterfactual definition [46], we define the Probability of Necessary and Sufficient Cause (PNS) for the step* $\mathbf{s}_t$ *as:*

$$\mathrm{PNS}(\mathbf{S}, \bar{\mathbf{s}}_t, \mathbf{q}) := P\left(\mathbf{A_S} = \mathbf{y}, \mathbf{A_{S'}} \neq \mathbf{y}\right). \tag{4}$$

This quantifies the probability that step $\mathbf{s}_t$ is sufficient and necessary for the correct answer under a counterfactual. Identifiability results are in Appendix A.

# 4 Methodology - PNS Estimation for Improving Chain-of-Thought Reasoning

In practice, directly maximizing the full-chain Probability of Necessity and Sufficiency (PNS) for reconstructed CoTs is computationally intractable. To address this, we adopt a two-stage causal optimization strategy: we first enhance the chain-level Probability of Sufficiency (PS), followed by step-wise refinement via the node-level Probability of Necessity (PN). Guided by these criteria, we propose an iterative pruning framework—Algorithm 1—that removes and reorders steps to preserve only those that are both causally sufficient and necessary for producing the correct answer. The overall process is illustrated in Figure 2, which shows how initial CoTs are evaluated, intervened upon, and refined to produce minimal causal traces. These optimized CoTs are then used as high-quality exemplars for subsequent in-context learning and fine-tuning, enabling the base model to internalize which reasoning paths are truly essential.

## 4.1 PNS Estimation and Algorithm for Reconstructing CoT

---
**Algorithm 1:** Sufficient and Necessary Optimization of CoT

---
**Input:** Initial CoT chain $\mathbf{S}_{\mathrm{init}}$, ground truth answer $\mathbf{y}$, query $\mathbf{q}$, threshold $\alpha$
**Output:** Optimized CoT chain $\mathbf{S}_{\mathrm{final}}$
$\hat{\mathbf{y}}_{\mathrm{init}} \leftarrow \mathrm{Rollout}(\mathbf{S}_{\mathrm{init}}, \mathbf{q})$;
$\mathrm{PS} \leftarrow \mathbb{1}[\![\hat{\mathbf{y}}_{\mathrm{init}} = \mathbf{y}]\!]$;
**if** $\mathrm{PS} = 1$ **then**
  Let $\mathbf{S}_{\mathrm{current}} \leftarrow \mathbf{S}_{\mathrm{init}}$;
  **foreach** *step* $\mathbf{s}_t \in \mathbf{S}_{\mathrm{current}}$ *(processed in order)* **do**
    $\bar{\mathbf{s}}_t \leftarrow \mathrm{GenerateAlternative}(\mathbf{s}^{\mathrm{current}}_{<t}, \mathbf{s}_t)$;
    $V_{\mathrm{scores}} \leftarrow$ empty list;
    **for** $j \leftarrow 1 \ldots k$ **do**
      $\overline{\mathbf{S}}^{(j)} \leftarrow \mathrm{RolloutContinuation}(\mathbf{s}^{\mathrm{current}}_{<t}, \bar{\mathbf{s}}_t, B)$;  // $B$ is the rollout model;
       Forms $(\mathbf{s}^{\mathrm{current}}_{<t}, \bar{\mathbf{s}}_t, \mathbf{s}'^{(j)}_{>t})$
      Ensure semantic disjointness of $(\bar{\mathbf{s}}_t, \mathbf{s}'^{(j)}_{>t})$ from original $(\mathbf{s}_t, \mathbf{s}^{\mathrm{current}}_{>t})$;
      Add $V(\overline{\mathbf{S}}^{(j)})$ to $V_{\mathrm{scores}}$ ;                  // $V$ is the validation model
    $\mathrm{PNS}_{\mathrm{val}}(\mathbf{s}_t) \leftarrow 1 - \frac{1}{k} \sum_{v \in V_{\mathrm{scores}}} v$;
    **if** $\mathrm{PNS}_{val}(\mathbf{s}_t) > \alpha$ **then**
      Append $\mathbf{s}_t$ to $\mathbf{S}_{\mathrm{final}}$ ;          // If $\mathbf{s}_t$ is deemed necessary, keep it
    **else**
      Skip $\mathbf{s}_t$ ;                              // Drop unnecessary step
**else**
  $\mathbf{S}_{\mathrm{final}} \leftarrow \mathbf{S}_{\mathrm{init}}$ ;                    // Original chain not sufficient
**return** $\mathbf{S}_{\mathrm{final}}$;

---

**Estimating PS (Chain-Level).** We approximate CoT trace sufficiency as binary: $\mathrm{PS} = 1$ if chain $\mathbf{S}$ yields the correct answer (equivalent to $P(\mathbf{A} = \mathbf{y} \mid \mathrm{do}(\mathbf{S}), \mathbf{q}) = 1$; Appendix A), else $\mathrm{PS} = 0$. To improve PS, we repeatedly execute Algorithm1 under the same question context. In each execution, the model samples an alternative CoT, and its PS is re-evaluated. This repeated sampling increases the likelihood of obtaining a CoT with higher PS. Lemma 1 (proof in Appendix A) establishes PNS identifiability (Definition 3.2) when $\mathrm{PS} = 1$:

**Lemma 1 (Identifiability of PNS under $P(\mathbf{A} = \mathbf{y} \mid \mathrm{do}(\mathbf{S}), \mathbf{q}) = 1$)** *Assume:*

1. *Perfect intervention with correct CoT* $\mathbf{S} = (\mathbf{s_{<t}}, \mathbf{s_t}, \mathbf{s_{>t}})$ *yields* $P(\mathbf{A} = \mathbf{y} \mid \mathrm{do}(\mathbf{S}), \mathbf{q}) = 1$.

2. *Replacing step* $\mathbf{s}_t$ *with incorrect* $\bar{\mathbf{s}}_t$ *and performing rollout* $\mathbf{s}'_{>t}$ *(from* $\bar{\mathbf{s}}_t$*) yields intervened chain* $\overline{\mathbf{S}} = (\mathbf{s}_{<t}, \bar{\mathbf{s}}_t, \mathbf{s}'_{>t})$.

*Then, PNS(*$\mathbf{S}, \bar{\mathbf{s}}_t, \mathbf{q}$*)* $= P(\mathbf{A}_{\mathrm{do}(\mathbf{S})} = \mathbf{y}, \mathbf{A}_{\mathrm{do}(\overline{\mathbf{S}})} \neq \mathbf{y} \mid \mathbf{q})$ *simplifies to* $1 - P(\mathbf{A} = \mathbf{y} \mid \mathrm{do}(\overline{\mathbf{S}}), \mathbf{q})$, *assuming perfect intervention and the nature of* $\overline{\mathbf{S}}$ *from Definition 3.2.*

When PS $= 1$, PNS validity depends on $P(\mathbf{A} = \mathbf{y} \mid \mathrm{do}(\bar{\mathbf{S}}), \mathbf{q})$, reflecting PN's magnitude.

**Estimating PN (Node-Level).** If $\mathbf{S}$ is sufficient (PS$(\mathbf{S}, \mathbf{q}) = 1$), Lemma 1 guides PN estimation for each node $\mathbf{s}_t$ to evaluate PNS$(\mathbf{S}, \bar{\mathbf{s}}_t, \mathbf{q})$. We construct $\bar{\mathbf{s}}_t$ by removing $\mathbf{s}_t$'s content and descendants. A rollout model $B$ generates a revised downstream segment $(\bar{\mathbf{s}}_t, \mathbf{s}'_{>t})$, forming intervened chain $\overline{\mathbf{S}}^{(i)} = (\mathbf{s}_{<t}, \bar{\mathbf{s}}_t, \mathbf{s}'^{(i)}_{>t})$ for the $i$-th rollout, ensuring $\bar{\mathbf{s}}_t, \mathbf{s}'^{(i)}_{>t}$ are semantically disjoint from original components rooted at $\mathbf{s}_t$. Each $\overline{\mathbf{S}}^{(i)}$ is assessed by validation model $V$ for coherence and logical integrity (beyond just final answer correctness). PNS is then computed via Monte-Carlo estimation over $k$ rollouts:

$$\mathrm{PNS}(\mathbf{S}, \bar{\mathbf{s}}_t, \mathbf{q}) \approx 1 - \frac{1}{k} \sum_{i=1}^{k} V(\overline{\mathbf{S}}^{(i)}). \tag{5}$$

Nodes with PNS score below threshold $\alpha$ (and their downstream nodes) are pruned iteratively until all retained nodes satisfy necessity.

**Iterative Optimization.** Algorithm 1 details the iterative optimization procedure. Starting from the initial CoT trace, we extract its chain and compute PNS. If the chain is sufficient, we perform necessity estimation for each node and prune the chain accordingly. The final optimized trace $\mathbf{S}_{\mathrm{final}}$ consists only of reasoning steps that are both sufficient to produce the correct answer and necessary to preserve logical coherence.

**Rollout Strategies for Intervention Chain $\overline{\mathbf{S}}^{(i)}$.** We use three strategies (details/prompts in Appendix B) for generating semantically modified steps for $\overline{\mathbf{S}}^{(i)}$: (1) *Direct Rollout*: base model generates replacement from preceding context. (2) *Prompt-Based Rollout*: structured prompts guide base model substitutions. (3) *External Rollout*: a separate, stronger model generates replacements. Base/rollout models are consistent for (1)-(2); external rollouts use a more capable auxiliary.

### 4.2 PNS-Guided Reasoning Enhancement for In-Context Learning and Fine-Tuning

We use causally filtered CoT traces to improve LLMs under two paradigms: **In-Context Learning (ICL)** and **Supervised Fine-Tuning (SFT)**. In ICL, optimized CoTs are directly inserted into prompts to guide non-reasoning models. In SFT, we fine-tune reasoning-capable models using 1,229 high-quality CoTs. Results for both settings are reported in § 5.2.

## 5 Experiments

Our experiments are structured around two core questions:

**RQ1:** Can our method construct CoT datasets that enhance causal sufficiency and necessity? (§ 5.2.1)

**RQ2:** Can the causally optimized CoT data improve the performance of non-reasoning models via ICL, and further enhance reasoning-capable models through SFT? (§ 5.2.2)

### 5.1 Experimental Setup

**Datasets.** We evaluate on diverse reasoning benchmarks to ensure robustness across domains and difficulty levels. For mathematical reasoning, we use: (1) **GSM-8k** [10], with grade-school problems; (2) **MATH-500** [25], covering intermediate-level topics; and (3) **AIME**, with advanced competition problems up to 2025 [44, 11]. For commonsense reasoning, we use **CommonsenseQA** [53], a multiple-choice dataset requiring everyday inference.

Table 1: Experimental results for RQ1. Comparison of CoT reasoning performance before and after PNS-based optimization across QWEN and DEEPSEEK variants.

| Method | GSM-8k | | | CommonsenseQA | | | MATH-500 | | | AIME | | |
|---|---|---|---|---|---|---|---|---|---|---|---|---|
| | Tokens (Initial/Final) | Steps (Initial/Final) | Acc. (Initial/Final) | Tokens (Initial/Final) | Steps (Initial/Final) | Acc. (Initial/Final) | Tokens (Initial/Final) | Steps (Initial/Final) | Acc. (Initial/Final) | Tokens (Initial/Final) | Steps (Initial/Final) | Acc. (Initial/Final) |
| Qwen Variant (QwQ-32B-Preview & Qwen-2.5-72B-Instruct) | | | | | | | | | | | | |
| Prompt-Based | 113.8 →33.9 | 8.1 →2.3 | 90.0% →95.8% | 109.2 →90.4 | 3.7 →3.0 | 69.7% →75.6% | 281.8 →178.8 | 9.2 →5.5 | 82.6% →86.6% | 531.3 →511.9 | 12.5 →12.3 | 16.7% →26.7% |
| Direct | 113.8 →26.6 | 8.1 →2.0 | 90.0% →97.0% | 109.2 →90.4 | 3.7 →3.0 | 69.7% →74.5% | 281.8 →169.4 | 9.2 →5.1 | 82.6% →87.4% | 531.3 →522.7 | 12.5 →12.3 | 16.7% →23.3% |
| External | 356.4 →58.9 | 23.9 →3.0 | 93.3% →97.9% | 474.2 →215.9 | 17.8 →7.4 | 83.2% →88.0% | 743.2 →200.7 | 50.3 →11.0 | 87.6% →94.3% | 1719.4 →1479.4 | 108.7 →76.7 | 43.3% →56.7% |
| DeepSeek Variant (DeepSeek-R1 & DeepSeek-V3) | | | | | | | | | | | | |
| Prompt-Based | 137.3 →29.6 | 5.4 →1.4 | 95.0% →97.3% | 191.0 →70.9 | 6.2 →2.7 | 83.0% →85.3% | 387.6 →163.4 | 16.2 →6.2 | 85.9% →92.0% | 2438.8 →2195.4 | 120.3 →100.9 | 25.0% →25.0% |
| Direct | 137.3 →29.2 | 5.4 →1.3 | 95.0% →97.0% | 191.0 →69.8 | 6.2 →2.5 | 83.0% →85.3% | 387.6 →161.2 | 16.2 →5.9 | 85.9% →91.6% | 2438.8 →2082.6 | 120.3 →96.9 | 25.0% →25.0% |
| External | 451.6 →135.1 | 6.7 →3.2 | 99.0% →99.9% | 368.2 →167.9 | 7.6 →3.1 | 87.1% →94.9% | 828.8 →214.5 | 29.7 →6.1 | 93.2% →97.6% | 3052.5 →1639.3 | 157.8 →68.3 | 79.2% →92.6% |

**Evaluation Metrics.** We assess (i) reasoning efficiency—measured by token length and step length—and (ii) accuracy, comprising PS(chain) and final-answer accuracy. Token length is computed via space-delimited tokenization; step length counts steps separated by a double newline for consistency. Accuracy is the average fraction of correctly answered instances on the test set, i.e.,

$$\text{Accuracy} = \frac{\text{\# correctly answered instances}}{\text{\# instances in the test set}}.$$

**Baselines.** For RQ1, we compare unoptimized CoT traces from the base model with PNS-optimized versions (via § 4.1) in terms of reasoning efficiency, accuracy, and average PNS. A representative example is shown in Figure 3, with full results in Appendix G.

For RQ2, ICL baselines include:(1) **Standard**, few-shot with original (often redundant) CoTs; (2) **Fast-Solve**, concise yet complete reasoning; (3) **Reduction** [19], shortcut conclusions; (4) **CoD** [67], minimal key phrases; and (5) **Ours-ICL**, few-shot with PNS-optimized, causally essential steps. Details and prompts are in Appendix C. For SFT, we compare: **Original**, the base model; **Noncausal**, fine-tuned on raw CoTs; and **Causal**, fine-tuned on the same CoTs after PNS-based pruning.

**Models.** For RQ1, we use `Qwen-2.5-72B-Instruct` [55] as both the base and rollout model. In the **External** variant, `QwQ-32B-Preview` [56] is used as base, with rollout unchanged. For DeepSeek, `DeepSeek-V3` [16] serves as both base and rollout in standard settings, while the external variant uses `DeepSeek-R1` [17] as base and V3 as rollout. PNS evaluations share the same configuration.

RQ2 ICL experiments primarily use `Qwen-2.5-72B-Instruct`, with additional results from `Qwen-2.5-7B-Instruct` [55], `Llama-3.1-8B-Instruct` [1], and `DeepSeek-V3`. SFT experiments fine-tune `DeepSeek-R1-Distill-Qwen-1.5B`, `DeepScaleR-1.5B-Preview` [37], and `Phi-4-mini-reasoning` [66]. Training details are in Appendix D.

### 5.2 Main Results

#### 5.2.1 PNS Optimization on CoT Trajectories

We apply our method to CoT traces from Qwen and DeepSeek variants. As shown in Table 1, our PNS-based algorithm reduces both token and step lengths while improving accuracy [4], indicating effective removal of redundant reasoning.

We compare average PNS values before and after optimization across tasks (AIME and CommonsenseQA) and models (`Qwen-2.5-72B-Instruct`, `DeepSeek-R1`). As shown in Figure 3, PNS values consistently increase after optimization, confirming that the retained steps are more causally sufficient and necessary. The figure illustrates results on 15 sampled questions per setting; more comprehensive results across larger test sets are provided in Appendix G.

---

[4]Inference for reasoning models was performed using VLLM. The `max-tokens` is 16,384.

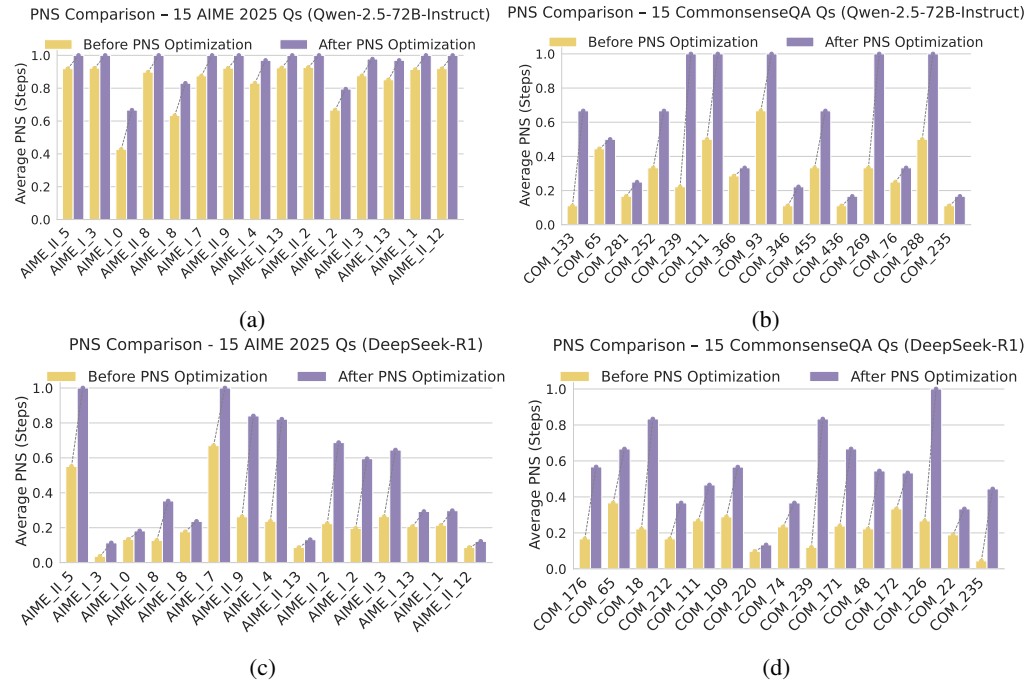

Figure 3: Average PNS values before and after optimization across different models and datasets. Each subfigure displays PNS improvements across 15 sampled problems: **(a)** `Qwen-2.5-72B-Instruct` evaluated on AIME, **(b)** `Qwen-2.5-72B-Instruct` on CommonsenseQA, **(c)** `DeepSeek-R1` evaluated on AIME, and **(d)** `DeepSeek-R1` on CommonsenseQA. PNS-optimization CoTs exhibit consistently higher PNS values, indicating an increased necessity for retained steps.

We conducted a human quality evaluation of 50 CoTs: 84% were judged both sufficient and necessary (S&N), and only 6% insufficient (NbI); see Appendix I for details.

These findings indicate that our optimized CoTs are not only more concise and accurate, but also demonstrate enhanced causal sufficiency and necessity. Notably, the average PNS per step increases after optimization, suggesting that the retained reasoning steps are more integral to producing correct answers—each step contributes more critically to the final outcome than before.

### 5.2.2 Enhancing LLMs via ICL and SFT with Optimized CoT

We investigate enhancing LLM performance with optimized CoT data via in-context learning (ICL) and supervised fine-tuning (SFT).

**In-Context Learning with Optimized CoT.** Using optimized CoT traces for ICL with non-fine-tuned LLMs (Table 2), Ours-ICL balances reasoning efficiency and accuracy. Compared to Standard CoT, it consistently reduces token/step usage (often >50%) with minimal/no accuracy loss. For instance, on GSM-8k with `DeepSeek-V3`, Ours-ICL improves accuracy (97.6% to 99.9%) while cutting tokens by 67%; with `Llama-3.1-8B-Instruct` on MATH-500, accuracy increases by 7.1 points (to 54.8%) with more concise reasoning. Ours-ICL also surpasses baselines like Fast-Solve and Reduction [19] in accuracy with comparable or better efficiency. On GSM-8k with `Qwen-2.5-72B-Instruct`, our method (99.5%, 65.3 tokens) outperforms Fast-Solve (91.7%, 72.8 tokens) and Reduction (84.7%, 114.1 tokens). Unlike aggressive pruning (e.g., CoD [67]), Ours-ICL maintains substantially higher accuracy, especially on complex tasks (e.g., MATH-500: 96.2% vs. 55.6% with `DeepSeek-V3`).

**Limitations for ICL.** Despite its effectiveness with non-reasoning models, ICL using optimized CoT is sensitive to prompt/example selection. Its benefits diminish on complex tasks (e.g., MATH-500), and performance is constrained by the fixed parameters of such models. In contrast, SFT is more impactful for reasoning-capable models, allowing deeper integration of reasoning patterns.

Table 2: Experimental results for RQ2 (ICL on Non-Reasoning Models). Lower is better (↓) for Tokens/Steps, higher is better (↑) for Acc. Change rates (%) relative to the average of methods for that metric/dataset/model are in parentheses. Cells significantly better than average are colored blue (deeper for greater improvement).

| Method | CommonsenseQA | | | GSM-8k | | | MATH-500 | | |
|---|---|---|---|---|---|---|---|---|---|
| | Tokens↓ | Steps↓ | Acc.↑ | Tokens↓ | Steps↓ | Acc.↑ | Tokens↓ | Steps↓ | Acc.↑ |
| | | | | | DeepSeek-V3 | | | | |
| Standard | 177.5 (+90.9%) | 5.7 (+54.1%) | 83.8% (+1.5%) | 157.3 (+83.3%) | 7.4 (+51.0%) | 97.6% (+3.0%) | 598.6 (+90.4%) | 26.7 (+88.0%) | 93.2% (+9.9%) |
| Fast-Solve | 120.1 (+29.1%) | 4.7 (+27.0%) | 82.0% (-0.7%) | 86.3 (+0.6%) | 4.9 (0.0%) | 95.1% (+0.3%) | 329.2 (+4.7%) | 13.6 (-4.2%) | 87.2% (+2.8%) |
| Reduction [19] | 103.4 (+11.2%) | 3.6 (-2.7%) | 83.1% (+0.6%) | 104.4 (+21.7%) | 6.0 (+22.4%) | 97.3% (+2.6%) | 476.3 (+51.5%) | 22.0 (+54.9%) | 91.6% (+8.0%) |
| CoD [67] | 19.3 (-79.2%) | 2.0 (-45.9%) | 80.7% (-2.3%) | 28.7 (-66.5%) | 1.8 (-63.3%) | 84.0% (-11.4%) | 31.2 (-90.1%) | 2.1 (-85.2%) | 55.6% (-34.4%) |
| Ours-ICL | 44.7 (-51.9%) | 2.7 (-27.0%) | 83.6% (+1.2%) | 52.2 (-39.2%) | 4.3 (-12.2%) | 99.9% (+5.4%) | 136.7 (-56.5%) | 6.4 (-54.9%) | 96.2% (+13.4%) |
| | | | | | Qwen-2.5-72B-Instruct | | | | |
| Standard CoT | 109.1 (+43.0%) | 3.7 (+27.6%) | 78.2% (-1.1%) | 113.8 (+47.8%) | 8.0 (+81.8%) | 93.6% (+4.6%) | 281.8 (+51.9%) | 9.2 (+27.8%) | 84.0% (+16.8%) |
| Fast-Solve | 59.7 (-21.7%) | 2.0 (-31.0%) | 67.0% (-15.3%) | 72.8 (-5.5%) | 2.7 (-38.6%) | 91.7% (+2.5%) | 192.4 (+3.7%) | 7.3 (+1.4%) | 69.8% (-2.9%) |
| Reduction [19] | 116.6 (+52.8%) | 3.5 (+20.7%) | 84.9% (+7.3%) | 114.1 (+48.2%) | 4.8 (-9.1%) | 84.7% (-5.4%) | 233.3 (+25.8%) | 9.4 (+30.6%) | 72.4% (+0.7%) |
| CoD [67] | 14.4 (-81.1%) | 2.1 (-27.6%) | 82.3% (+4.0%) | 18.8 (-75.6%) | 1.1 (-74.9%) | 78.1% (-12.7%) | 23.0 (-87.6%) | 1.2 (-83.3%) | 52.0% (-27.6%) |
| Ours-ICL | 81.6 (+7.0%) | 3.4 (+17.2%) | 83.0% (+4.9%) | 65.3 (-15.2%) | 5.3 (+20.5%) | 99.5% (+11.2%) | 196.9 (+6.1%) | 8.9 (+23.6%) | 81.2% (+13.0%) |
| | | | | | Qwen-2.5-7B-Instruct | | | | |
| Standard CoT | 209.8 (+68.0%) | 7.2 (+46.9%) | 70.4% (-6.5%) | 149.7 (+48.8%) | 7.2 (+33.3%) | 85.1% (+0.6%) | 263.5 (+43.0%) | 9.7 (+19.8%) | 71.0% (+12.9%) |
| Fast-Solve | 120.1 (-3.8%) | 5.0 (+2.0%) | 76.5% (+1.6%) | 108.4 (+7.8%) | 5.5 (+1.9%) | 83.6% (-1.2%) | 200.6 (+8.8%) | 8.4 (+3.7%) | 64.6% (+2.7%) |
| Reduction [19] | 178.5 (+42.9%) | 6.8 (+38.8%) | 74.4% (-1.2%) | 131.1 (+30.3%) | 6.4 (+18.5%) | 84.3% (-0.4%) | 232.2 (+26.0%) | 9.1 (-2.4%) | 71.8% (+14.1%) |
| CoD [67] | 16.8 (-86.5%) | 1.9 (-60.7%) | 77.4% (+2.8%) | 30.3 (-69.9%) | 2.9 (-46.3%) | 75.7% (-10.5%) | 50.3 (-72.7%) | 5.8 (-28.3%) | 34.6% (-45.0%) |
| Ours-ICL | 99.1 (-20.7%) | 3.8 (-22.4%) | 77.6% (+3.1%) | 83.4 (-17.1%) | 4.8 (-11.1%) | 94.1% (+11.2%) | 174.7 (-5.2%) | 7.7 (-4.9%) | 72.6% (+15.4%) |
| | | | | | Llama-3.1-8B-Instruct | | | | |
| Standard CoT | 169.3 (+33.5%) | 7.3 (+15.9%) | 72.2% (+2.8%) | 182.8 (+29.2%) | 7.9 (-1.2%) | 79.2% (-0.1%) | 741.3 (+55.2%) | 46.0 (+39.8%) | 47.6% (+6.7%) |
| Fast-Solve | 140.7 (+10.9%) | 7.9 (+25.4%) | 69.0% (-1.7%) | 170.6 (+20.6%) | 9.5 (+18.2%) | 72.0% (-9.2%) | 453.1 (-5.1%) | 30.9 (-6.1%) | 46.4% (+4.0%) |
| Reduction [19] | 143.2 (+13.0%) | 5.8 (-7.9%) | 69.9% (-0.4%) | 129.1 (-8.8%) | 6.2 (-22.4%) | 82.8% (+4.4%) | 515.6 (+8.0%) | 34.0 (+3.3%) | 46.2% (+3.6%) |
| CoD [67] | 60.6 (-52.2%) | 3.5 (-44.3%) | 67.6% (-3.7%) | 96.7 (-31.6%) | 8.5 (+6.2%) | 69.2% (-12.7%) | 312.5 (-34.6%) | 29.2 (-11.2%) | 28.0% (-37.2%) |
| Ours-ICL | 120.3 (-5.1%) | 7.1 (+12.7%) | 72.1% (+2.7%) | 128.3 (-9.3%) | 8.1 (+1.2%) | 93.1% (+17.4%) | 365.9 (-23.4%) | 24.2 (-26.4%) | 54.8% (+22.9%) |

**Supervised Fine-Tuning with Optimized CoT.** We fine-tune reasoning models on 1,229 PNS-selected CoT traces from MATH [25], MMLU [24], ZebraLogicBench [34], CommonsenseQA [53], and AIME (pre-2024) [44]. All traces were manually checked for causal sufficiency, necessity, concision, and quality; CoTs selected by our method were not edited—only verified. Table 3 demonstrates Causal-CoT's consistent outperformance over baselines. On CommonsenseQA, it improves accuracy to 47.2% (from 37.6% on `DeepSeek-R1-Distill-Qwen-1.5B`, and from 41.3% on `DeepScaleR-1.5B-Preview`) while halving reasoning steps. On MATH-500, it achieves 78.2% accuracy (33.1 steps) versus the Original model's 76.4% (77.2 steps). Even on difficult tasks like AIME25 (low absolute performance[5]), Causal-CoT significantly cuts reasoning length (e.g., `DeepSeek-R1-Distill-Qwen-1.5B`: 212.4 to 95.4 tokens). The Causal variant also matches/exceeds Noncausal fine-tuning with substantially fewer steps/tokens. For GSM-8k, it reaches 86.2% accuracy (11.6 steps), while Noncausal needs 15 steps for 86.1% (`DeepScaleR-1.5B-Preview`).

**Implications of SFT Results.** SFT on PNS-selected traces yields consistent gains in accuracy and reasoning efficiency, even with small Causal-CoT datasets, confirming the high supervision value of enforcing causal sufficiency and necessity. The token overhead from PNS filtering is a one-time curation cost; after fine-tuning, inference becomes cheaper because the model generates concise, accurate CoTs without stepwise rollouts or post-hoc pruning. The strength of our approach lies in reshaping the training distribution toward high-PNS evidence, enabling the model to internalize causally meaningful, non-redundant reasoning patterns. At test time, this manifests as streamlined, interpretable CoTs that improve both efficiency and reliability.

# 6  Conclusion

This work successfully incorporates PNS into CoT reasoning. The developed method systematically prunes unnecessary reasoning steps, leading to significant improvements in reasoning efficiency, while maintaining or even enhancing the accuracy of the outcomes. The effectiveness of the method has been verified across both in-context learning and supervised fine-tuing scenarios.

**Limitation and Future Work.** Limitations include potential performance decreases on highly complex tasks. Key challenges involve selecting optimal pruning thresholds, managing PNS esti-

---

[5]Inference performed via VLLM; The `max-tokens` is 16,384.

Table 3: Experimental results for RQ2 (SFT on Reasoning Models). For Noncausal and Causal methods, change rates (%) are reported relative to the "Original" for each model/dataset. Cells with notable improvements over the original are highlighted.

| Method | CommonsenseQA | | | GSM-8k | | | MATH-500 | | | AIME25 | | |
|---|---|---|---|---|---|---|---|---|---|---|---|---|
| | Tokens↓ | Steps↓ | Acc.↑ | Tokens↓ | Steps↓ | Acc.↑ | Tokens↓ | Steps↓ | Acc.↑ | Tokens↓ | Steps↓ | Acc.↑ |
| DeepSeek-R1-Distill-Qwen-1.5B | | | | | | | | | | | | |
| Original | 751.3 | 21.4 | 37.6% | 332.1 | 14.4 | 77.9% | 1441.8 | 77.2 | 76.4% | 4002.3 | 212.4 | 23.3% |
| | *(baseline)* | *(baseline)* | *(baseline)* | *(baseline)* | *(baseline)* | *(baseline)* | *(baseline)* | *(baseline)* | *(baseline)* | *(baseline)* | *(baseline)* | *(baseline)* |
| Noncausal | 1271.3 | 27.9 | **43.2%** | 621.4 | 16.5 | **83.4%** | 1456.6 | **54.7** | **81.6%** | 3796.3 | **116.4** | 20.0% |
| | (+69.2%) | (+30.4%) | (+14.9%) | (+87.1%) | (+14.6%) | (+7.1%) | (+1.0%) | (-29.1%) | (+6.8%) | (-5.1%) | (-45.2%) | (-14.2%) |
| Causal (Ours) | 740.0 | **10.3** | **47.2%** | 327.8 | 12.8 | **84.2%** | **911.9** | **33.1** | 78.2% | 2948.3 | 95.4 | 23.3% |
| | (-1.5%) | (-51.9%) | (+25.5%) | (-1.3%) | (-11.1%) | (+8.1%) | (-36.7%) | (-57.1%) | (+2.4%) | (-26.3%) | (-55.1%) | (0.0%) |
| DeepScaleR-1.5B-Preview | | | | | | | | | | | | |
| Original | 646.7 | 22.1 | 41.3% | 716.3 | 27.5 | 87.5% | 1325.0 | 77.4 | 84.6% | 4897.2 | 356.0 | 20.0% |
| | *(baseline)* | *(baseline)* | *(baseline)* | *(baseline)* | *(baseline)* | *(baseline)* | *(baseline)* | *(baseline)* | *(baseline)* | *(baseline)* | *(baseline)* | *(baseline)* |
| Noncausal | 611.5 | 21.7 | 40.8% | **604.8** | **15.0** | 86.1% | 1445.5 | **52.4** | 82.8% | 5709.8 | **243.2** | 16.7% |
| | (-5.4%) | (-1.8%) | (-1.2%) | (-15.6%) | (-45.5%) | (-1.6%) | (+9.1%) | (-32.3%) | (-2.1%) | (+16.6%) | (-31.7%) | (-16.5%) |
| Causal (Ours) | 601.2 | **11.2** | **47.2%** | 394.6 | **11.6** | 86.2% | **1041.4** | **36.3** | **91.7%** | 2015.4 | **53.6** | 20.0% |
| | (-7.0%) | (-49.3%) | (+14.3%) | (-44.9%) | (-57.8%) | (-1.5%) | (-21.4%) | (-53.1%) | (+8.4%) | (-58.8%) | (-85.0%) | (0.0%) |
| Phi-4-mini-reasoning | | | | | | | | | | | | |
| Original | 935.7 | 18.6 | 72.4% | 783.3 | 18.6 | 92.6% | 1743.5 | 62.1 | 85.8% | 6544.0 | 274.3 | 30.0% |
| | *(baseline)* | *(baseline)* | *(baseline)* | *(baseline)* | *(baseline)* | *(baseline)* | *(baseline)* | *(baseline)* | *(baseline)* | *(baseline)* | *(baseline)* | *(baseline)* |
| Noncausal | 949.6 | **14.9** | 66.4% | **566.9** | **14.3** | 89.8% | 2042.8 | 65.8 | 59.8% | 8297.8 | 359.3 | 23.3% |
| | (+1.5%) | (-19.9%) | (-8.3%) | (-27.6%) | (-23.1%) | (-3.0%) | (+17.2%) | (+5.9%) | (-30.3%) | (+26.8%) | (+31.0%) | (-22.3%) |
| Causal (Ours) | 920.3 | **14.1** | **72.9%** | 517.2 | **14.0** | 92.4% | **1031.0** | **28.7** | 86.7% | **4140.0** | **112.0** | 30.0% |
| | (-1.6%) | (-24.2%) | (+0.7%) | (-34.0%) | (-24.7%) | (-0.2%) | (-40.9%) | (-53.8%) | (+1.0%) | (-36.7%) | (-59.2%) | (0.0%) |

mation costs, and ensuring counterfactual generation quality. Future work will aim to address these limitations, focusing on improving the causal fidelity and overall performance of LLM reasoning.

## Acknowledgments

X. Yu was supported in part by the National Natural Science Foundation of China (Nos. 62472306, 62441221, and 62206116), Tianjin University's 2024 Special Project on Disciplinary Development (No. XKJS-2024-5-9), the Tianjin University Talent Innovation Reward Program for Literature & Science Graduate Students (No. C1-2022-010), and the Henan Province Key Research and Development Program (No. 251111210500). H. Li was supported in part by National Natural Science Foundation of China (No. 623B2002).

## Author Contributions

This work was completed through the joint efforts of all authors. X.Y and Z.W were responsible for the main experiment design, model implementation, and manuscript writing. M.Y contribute to idea formulate, experiment design, theoretical analysis and manuscript writing. L.Y contribute to experimental design. H.L contribute to idea formulate and manuscript writing. The other authors contributed to regular discussion.

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

# Appendix

## Appendix Table of Contents

---

# A   Theoretical Analysis

## A.1   Exogeneity and Monotonicity Conditions

For the calculation of the Probability of Necessary and Sufficient (PNS) conditions, it is essential that the exogeneity and monotonicity conditions are satisfied.

**Definition 5 (CoT Exogeneity)** *When generating each reasoning step* $\mathbf{s_t}$, *the generation of the current step depends only on the previous reasoning steps* $\mathbf{s_{<t}}$ *and the question* $\mathbf{q}$, *and not on any external variables. Formally:*

$$P(\mathbf{s_t}|\operatorname{do}(\mathbf{s_{<t}}), \mathbf{q}) = P(\mathbf{s_t}|\mathbf{s_{<t}}, \mathbf{q}) \tag{A.1}$$

Where $\mathbf{s_t}$ is the current reasoning step, $\mathbf{s_{<t}}$ denotes all previous reasoning steps, and $\mathbf{q}$ is the input question. According to the definition of exogeneity, the intervention probability can be evaluated by the conditional probability.

**Assumption 1 (Monotonicity for CoT)** *Inspired by the monotonicity assumption in causal inference [46], we define monotonicity for Chain-of-Thought (CoT) reasoning as follows. Let* $\mathbf{S} = (\mathbf{s_1}, \ldots, \mathbf{s_n})$ *be a reasoning chain leading to the correct answer* $\mathbf{A} = \mathbf{y}$, *and let* $\overline{\mathbf{S}} = (\mathbf{s_1}, \ldots, \mathbf{s_{t-1}}, \overline{\mathbf{s}}_t, \mathbf{s'_{t+1}}, \ldots, \mathbf{s'_n})$ *denote a modified reasoning chain where only step* $t$ *is altered to an incorrect step* $\overline{\mathbf{s}}_t$ *and subsequent steps* $\mathbf{s'_{>t}}$ *are rolled out. CoT reasoning satisfies monotonicity if and only if, for every reasoning step* $t$ *and for every possible alteration* $\overline{\mathbf{s}}_t$, *the following holds:*

$$P\left(\mathbf{A_S} \neq \mathbf{y}, \mathbf{A_{\overline{S}}} = \mathbf{y}\right) = 0.$$

This condition states that altering a reasoning step to an incorrect version cannot result in correcting a previously incorrect final answer. Equivalently, whenever the modified chain $\overline{\mathbf{S}}$ yields a correct answer, the original unaltered chain $\mathbf{S}$ must also yield a correct answer, ensuring monotonic progression toward correctness.

## A.2   Identifiability of PNS in CoT Under Monotonicity Assumption

**Lemma 2 (Identifiability of PNS under downstream-adaptive reasoning)** *Assume that the Chain-of-Thought (CoT) reasoning process satisfies both the Exogeneity (Definition A.1) and Monotonicity (Assumption 1) assumptions. Let the correct reasoning chain be denoted by* $\mathbf{S} = (\mathbf{s_{<t}}, \mathbf{s_t}, \mathbf{s_{>t}})$, *and let an alternative corrupted step* $\overline{\mathbf{s}}_t$ *induce a modified future reasoning sequence* $\mathbf{s'_{>t}}$, *resulting in the altered chain* $\mathbf{S'} = (\mathbf{s_{<t}}, \overline{\mathbf{s}}_t, \mathbf{s'_{>t}})$. *Then the Probability of Necessary and Sufficient Cause (PNS) for the reasoning chain is identifiable and satisfies:*

$$\text{PNS}(\mathbf{S}, \overline{\mathbf{s}}_t) = P(\mathbf{A_S} = \mathbf{y}) - P(\mathbf{A_{S'}} = \mathbf{y}) = P(\mathbf{A} = \mathbf{y} \mid \operatorname{do}(\mathbf{S})) - P(\mathbf{A} = \mathbf{y} \mid \operatorname{do}(\mathbf{S'})).$$

*Note:* $\operatorname{do}(\mathbf{S'})$ *is equivalent to* $\operatorname{do}(\mathbf{s_{<t}}, \overline{\mathbf{s}}_t, \mathbf{s'_{>t}})$.

We begin with the definition of PNS as a counterfactual joint:

$$\text{PNS}(\mathbf{S}, \bar{\mathbf{s}}_t) = P\left(\mathbf{A}_\mathbf{S} = \mathbf{y}, \mathbf{A}_{\mathbf{S}'} \neq \mathbf{y}\right).$$

Using the fact that $\mathbf{A}_\mathbf{S} = \mathbf{y}$ implies either $\mathbf{A}_{\mathbf{S}'} = \mathbf{y}$ or $\mathbf{A}_{\mathbf{S}'} \neq \mathbf{y}$, we can rewrite:

$$P(\mathbf{A}_\mathbf{S} = y) = P(\mathbf{A}_\mathbf{S} = \mathbf{y}, \mathbf{A}_{\mathbf{S}'} = \mathbf{y}) + P(\mathbf{A}_\mathbf{S} = \mathbf{y}, \mathbf{A}_{\mathbf{S}'} \neq \mathbf{y}).$$

Thus:

$$\text{PNS}(\mathbf{S}, \bar{\mathbf{s}}_t) = P(\mathbf{A}_\mathbf{S} = \mathbf{y}) - P(\mathbf{A}_\mathbf{S} = \mathbf{y}, \mathbf{A}_{\mathbf{S}'} = \mathbf{y}).$$

Under the Monotonicity assumption (Assumption 1), $\mathbf{A}_{\mathbf{S}'} = \mathbf{y} \Rightarrow \mathbf{A}_\mathbf{S} = \mathbf{y}$. This implies that the event $(\mathbf{A}_\mathbf{S} = \mathbf{y}$ and $\mathbf{A}_{\mathbf{S}'} = \mathbf{y})$ is equivalent to the event $(\mathbf{A}_{\mathbf{S}'} = \mathbf{y})$. Therefore:

$$P(\mathbf{A}_\mathbf{S} = \mathbf{y}, \mathbf{A}_{\mathbf{S}'} = \mathbf{y}) = P(\mathbf{A}_{\mathbf{S}'} = \mathbf{y}).$$

Hence:

$$\text{PNS}(\mathbf{S}, \bar{\mathbf{s}}_t) = P(\mathbf{A}_\mathbf{S} = \mathbf{y}) - P(\mathbf{A}_{\mathbf{S}'} = \mathbf{y}).$$

Finally, under the Exogeneity assumption (Definition A.1), these counterfactual probabilities can be identified with interventional probabilities:

$$\text{PNS}(\mathbf{S}, \bar{\mathbf{s}}_t) = P(\mathbf{A} = \mathbf{y} \mid \text{do}(\mathbf{S})) - P(\mathbf{A} = \mathbf{y} \mid \text{do}(\mathbf{S}')).$$

### A.3   Identifiability of PNS without Monotonicity Assumption

**Lemma 3 (Identifiability of PNS under $P(\mathbf{A} = \mathbf{y} \mid \text{do}(\mathbf{S})) = 1$ without Monotonicity)**
*Assume:*

1. *Under the perfect intervention of the correct CoT chain $\mathbf{S} = (\mathbf{s}_{<t}, \mathbf{s}_t, \mathbf{s}_{>t})$, the model always produces the correct answer:*

$$P\left(\mathbf{A} = \mathbf{y} \mid \text{do}(\mathbf{S})\right) = 1.$$

2. *Replacing step $t$ with an incorrect step $\bar{\mathbf{s}}_t$ and allowing arbitrary rollout continuation $\mathbf{s}'_{>t}$ yields the intervened chain $\mathbf{S}' = (\mathbf{s}_{<t}, \bar{\mathbf{s}}_t, \mathbf{s}'_{>t})$.*

3. *We do not assume Monotonicity (Assumption 1), i.e., we make no assumption that $P\left(\mathbf{A}_{\text{do}(\mathbf{S})} \neq \mathbf{y}, \mathbf{A}_{\text{do}(\mathbf{S}')} = \mathbf{y}\right) = 0$. (Note: original text had $\mathbf{A}_{\text{do}(\bar{\mathbf{s}}_\mathbf{t})}$ which seems less precise here than $\mathbf{A}_{\text{do}(\mathbf{S}')}$ for the full chain).*

*Then the counterfactual joint defining PNS,*

$$\text{PNS}(\mathbf{S}, \bar{\mathbf{s}}_t) = P\left(\mathbf{A}_{\text{do}(\mathbf{S})} = \mathbf{y}, \ \mathbf{A}_{\text{do}(\mathbf{S}')} \neq \mathbf{y}\right),$$

*is identifiable and simplifies to:*

$$\text{PNS}(\mathbf{S}, \bar{\mathbf{s}}_t) = 1 - P\left(\mathbf{A} = \mathbf{y} \mid \text{do}(\mathbf{S}')\right).$$

Start from the definition:

$$\text{PNS}(\mathbf{S}, \bar{\mathbf{s}}_t) = P\left(A_{\text{do}(\mathbf{S})} = \mathbf{y}, \ A_{\text{do}(\mathbf{S}')} \neq \mathbf{y}\right).$$

Apply the law of total probability to the event $\mathbf{A}_{\text{do}(\mathbf{S})} = \mathbf{y}$:

$$P\left(\mathbf{A}_{\text{do}(\mathbf{S})} = \mathbf{y}\right) = P\left(\mathbf{A}_{\text{do}(\mathbf{S})} = \mathbf{y}, \ \mathbf{A}_{\text{do}(\mathbf{S}')} \neq \mathbf{y}\right) + P\left(\mathbf{A}_{\text{do}(\mathbf{S})} = \mathbf{y}, \ \mathbf{A}_{\text{do}(\mathbf{S}')} = \mathbf{y}\right).$$

Since $P(\mathbf{A} = \mathbf{y} \mid \text{do}(\mathbf{S})) = 1$ by assumption 1, the left side equals 1, so

$$1 = \text{PNS}(\mathbf{S}, \bar{\mathbf{s}}_t) + P\left(\mathbf{A}_{\text{do}(\mathbf{S}')} = \mathbf{y}, \ \mathbf{A}_{\text{do}(\mathbf{S})} = \mathbf{y}\right).$$

Under Exogeneity (Definition A.1, implying no hidden confounding between the choice of intervention and its outcome), observing the original chain $\mathbf{S}$ does not influence the outcome of intervening with $\mathbf{S}'$. Thus:

$$P\left(\mathbf{A}_{\text{do}(\mathbf{S}')} = \mathbf{y}, \ \mathbf{A}_{\text{do}(\mathbf{S})} = \mathbf{y}\right) = P\left(\mathbf{A}_{\text{do}(\mathbf{S}')} = \mathbf{y}\right) = P\left(\mathbf{A} = \mathbf{y} \mid \text{do}(\mathbf{S}')\right).$$

Substitute back to obtain

$$1 = \text{PNS}(\mathbf{S}, \bar{\mathbf{s}}_t) + P\left(\mathbf{A} = \mathbf{y} \mid \text{do}(\mathbf{S}')\right),$$

and therefore

$$\text{PNS}(\mathbf{S}, \bar{\mathbf{s}}_t) = 1 - P\left(\mathbf{A} = \mathbf{y} \mid \text{do}(\mathbf{S}')\right).$$

## A.4 Equivalence of Perfect Intervention and Full Sufficiency

**Theorem 1 (Equivalence of Perfect Intervention and Full Sufficiency)** *Under the standard Exogeneity assumption (Definition A.1), and assuming there exists at least one alternative chain $\overline{\mathbf{S}}$ (which could be the original chain before applying the correct steps $\mathbf{S}$, or any other relevant baseline) such that it has a nonzero probability of leading to failure, i.e., $P\big(\mathbf{A} \neq \mathbf{y}, \overline{\mathbf{S}}, \mathbf{q}\big) > 0$, the following two statements are equivalent:*

1. $P\big(\mathbf{A} = \mathbf{y} \mid \mathrm{do}(\mathbf{S}), \mathbf{q}\big) = 1$ *(Perfect Intervention with $\mathbf{S}$ guarantees $\mathbf{y}$).*

2. $\mathrm{PS}(\mathbf{S}, \mathbf{q}) = 1$, *where* $\mathrm{PS}(\mathbf{S}, \mathbf{q}) := P\big(\mathbf{A}_{\mathrm{do}(\mathbf{S})} = \mathbf{y} \mid \mathbf{A} \neq \mathbf{y}, \overline{\mathbf{S}}, \mathbf{q}\big)$ *(Full Sufficiency).*

($\Rightarrow$) Assume $P\big(\mathbf{A} = \mathbf{y} \mid \mathrm{do}(\mathbf{S}), \mathbf{q}\big) = 1$. This means the intervention $\mathrm{do}(\mathbf{S})$ guarantees $\mathbf{A} = \mathbf{y}$ in all worlds compatible with $\mathbf{q}$. Consider the definition of $\mathrm{PS}(\mathbf{S}, \mathbf{q})$:

$$\mathrm{PS}(\mathbf{S}, \mathbf{q}) = P\big(\mathbf{A}_{\mathrm{do}(\mathbf{S})} = \mathbf{y} \mid \mathbf{A} \neq \mathbf{y}, \overline{\mathbf{S}}, \mathbf{q}\big).$$

Given the condition $\mathbf{A} \neq \mathbf{y}, \overline{\mathbf{S}}, \mathbf{q}$, we evaluate the probability of $\mathbf{A}_{\mathrm{do}(\mathbf{S})} = \mathbf{y}$. Since $P\big(\mathbf{A} = \mathbf{y} \mid \mathrm{do}(\mathbf{S}), \mathbf{q}\big) = 1$, it follows that $\mathbf{A}_{\mathrm{do}(\mathbf{S})} = \mathbf{y}$ holds universally under $\mathrm{do}(\mathbf{S})$, including in those specific circumstances where $\mathbf{A} \neq \mathbf{y}$ would have occurred with $\overline{\mathbf{S}}$. Therefore, $P\big(\mathbf{A}_{\mathrm{do}(\mathbf{S})} = \mathbf{y} \mid \mathbf{A} \neq \mathbf{y}, \overline{\mathbf{S}}, \mathbf{q}\big) = 1$, so $\mathrm{PS}(\mathbf{S}, \mathbf{q}) = 1$.

($\Leftarrow$) Assume $\mathrm{PS}(\mathbf{S}, \mathbf{q}) = 1$. By definition, this means:

$$P\big(\mathbf{A}_{\mathrm{do}(\mathbf{S})} = \mathbf{y} \mid \mathbf{A} \neq \mathbf{y}, \overline{\mathbf{S}}, \mathbf{q}\big) = 1.$$

This implies that for any world compatible with $\mathbf{q}$ where $\mathbf{A} \neq \mathbf{y}$ would occur with $\overline{\mathbf{S}}$, applying $\mathrm{do}(\mathbf{S})$ results in $\mathbf{A} = \mathbf{y}$. We want to show $P\big(\mathbf{A} = \mathbf{y} \mid \mathrm{do}(\mathbf{S}), \mathbf{q}\big) = 1$, which is $P\big(\mathbf{A}_{\mathrm{do}(\mathbf{S})} = \mathbf{y} \mid \mathbf{q}\big) = 1$. Consider the outcome $A_{\mathrm{do}(\mathbf{S})}$ given $\mathbf{q}$. The intervention $\mathrm{do}(\mathbf{S})$ sets the chain of thought to $\mathbf{S}$ and determines the outcome $\mathbf{A}$. This outcome $\mathbf{A}_{\mathrm{do}(\mathbf{S})}$ is determined solely by $\mathbf{S}$ and $\mathbf{q}$ (due to exogeneity of $\mathbf{S}$ with respect to other factors once $\mathrm{do}(\mathbf{S})$ is applied). If $\mathrm{PS}(\mathbf{S}, \mathbf{q}) = 1$, it means $\mathrm{do}(\mathbf{S})$ corrects all instances where $\overline{\mathbf{S}}$ would lead to failure. What about instances where $\overline{\mathbf{S}}$ might lead to success ($\mathbf{A} = \mathbf{y}$)? Since $\mathbf{S}$ is the "correct" chain of thought designed to produce $\mathbf{y}$, the intervention $\mathrm{do}(\mathbf{S})$ is assumed to robustly produce $\mathbf{y}$. If it produces $\mathbf{y}$ when $\overline{\mathbf{S}}$ would have failed, and it (by its nature as a correct CoT) produces $\mathbf{y}$ when $\overline{\mathbf{S}}$ might have succeeded, then $\mathbf{A}_{\mathrm{do}(\mathbf{S})} = \mathbf{y}$ holds across all situations defined by $\mathbf{q}$ and any alternative $\overline{\mathbf{S}}$. Therefore, $P(\mathbf{A}_{\mathrm{do}(\mathbf{S})} = \mathbf{y} \mid \mathbf{q}) = 1$.

# B Intervention Prompts

**System Message for LLM Intervention**

> You are a helpful assistant. Continue solving the problem using mathematical expressions only, without repeating previous steps. Provide the final answer once, directly linked to the preceding reasoning, without additional summaries or explanations. Avoid using summarizing words such as 'so' or 'thus,' and refrain from repeating the final result when the calculation is already clear. Don't say something like "Let's continue with the previous reasoning" or other nonsense, just output the following reasoning directly.

**Direct and External Intervention Prompt**

> Question:
> {query}
> Current reasoning steps:
> {context_steps}

**Prompt-Based Intervention Prompt**

Ensure the next output node does not match the meaning of:
{current_step}
Avoid repeating the final result directly when the calculation is already clear.
Question:
{query}
Current reasoning steps:
{context_steps}

## C  In-Context Learning Prompts

**ICL Baselines.**    The ICL baselines consist of the following variants:

1. **Standard**: This baseline uses verbose Chain-of-Thought (CoT) exemplars that include all intermediate reasoning steps, regardless of redundancy. It reflects the default strategy often employed in prompting LLMs for step-by-step reasoning.

2. **Fast-Solve**: This baseline encourages the model to produce concise reasoning chains that contain only the minimal steps necessary to reach the correct answer, avoiding verbose or redundant elaboration.

3. **Reduction** [19]: This method emphasizes rapid completion by prompting models to directly output shortcut solutions, often skipping step-by-step logical progression. It reflects a minimalistic strategy that favors brevity over transparency.

4. **Chain-of-Draft (CoD)** [67]: This variant uses prompts composed of minimally informative intermediate phrases—enough to scaffold the reasoning process but without detailed elaboration—simulating a rough-draft-style reasoning chain.

5. **Ours-ICL**: Our method, which leverages traces optimized using the *Probability of Necessity and Sufficiency* (PNS), presents only causally essential reasoning steps. These exemplars are pruned to retain only those steps that significantly contribute to correct outcomes, ensuring both efficiency and fidelity in reasoning.

**System Message (Common for All Prompts)**

You are a helpful assistant who is good at reasoning. Whenever doing multistep reasoning, please use two newline characters to split multiple steps (\n\n).

**Ours-ICL Prompt (Sufficient and Necessary Reasoning)**

**User Message:**
**Instructions**
When solving the following questions, your reasoning should: - **Be Accurate:** Ensure your chain of thought leads to the correct answer without skipping any necessary logical steps.
- **Be Efficient:** Avoid unnecessary or redundant steps. Each step should be necessary to progress toward the solution.
- **Aim for Sufficient and Necessary Reasoning:** Only include steps that are both sufficient to reach the correct answer and necessary to avoid gaps or confusion. If a step can be removed without affecting correctness, remove it.
- **Notice the Pattern:** In the following examples, compare the original, verbose solution with the optimized solution. Learn to identify and eliminate redundant reasoning steps while preserving logical soundness.
—
**Example 1:** {example1}
**Example 2:** {example2}

> **Example 3:** {example3}
> **Now Solve This:**
> *Question:* {question}
> *Your Simplified and Optimized Answer:*

**Fast-Solve Prompt**

> **User Message:**
> You are a math assistant that solves problems step by step. Please reason in a clear and structured manner, but keep your explanation as concise as possible. Avoid unnecessary repetition or redundant steps. The goal is to solve the problem accurately with the fewest necessary steps.
> **Now Solve This:**
> *Question:* {question}
> *Your Simplified and Optimized Answer:*

**CoD Prompt**

> **User Message:**
> Think step by step, but only keep a minimum draft for each thinking step, with 5 words at most. Return the answer at the end of the response after a separator
> **Now Solve This:**
> *Question:* {question}
> *Your Simplified and Optimized Answer:*

**Reduction Prompt**

> **User Message:**
> Let's quickly conclude the answer with shortcut reasoning.
> **Now Solve This:**
> *Question:* {question}
> *Your Simplified and Optimized Answer:*

# D   Supervised Fine-Tuning (SFT) Hyperparameter Settings

Table 4 details the general hyperparameter configuration used for supervised fine-tuning (SFT) in our experiments. All SFT training was conducted on 8 NVIDIA RTX 3090 GPUs using the ZeRO-3 optimizer for efficient memory distribution. To accelerate training and reduce memory usage, we employed `bf16` mixed-precision computation.

The training used the `flash_attention_2` implementation for efficient attention computation, combined with a cosine learning rate scheduler that decays to a minimum learning rate. Each GPU was assigned a batch size of 1 due to the large context length of 16,384 tokens. The model was trained for 3 epochs, and `max_steps` was left as `-1` to allow epoch-based termination. These settings balance computational feasibility and performance under long-context, reasoning-intensive tasks.

The same configuration was applied across all target models, including `DeepSeek-R1-Qwen-1.5B`, `DeepScaleR-1.5B-Preview`, and `Phi-4-mini-reasoning`, unless otherwise specified.

Table 4: General SFT Hyperparameters. Hardware: $8 \times$ NVIDIA RTX 3090 GPUs, ZeRO-3 optimizer, bf16 mixed precision.

| Parameter | Value |
|---|---|
| attn_implementation | flash_attention_2 |
| bf16 | true |
| learning_rate | 5.0e-05 |
| lr_scheduler_type | cosine_with_min_lr |
| per_device_train_batch_size | 1 |
| max_steps | -1 |
| max_length | 16384 |
| num_train_epochs | 3 |

# E  In-Context Learning: Case study

To further illustrate the effectiveness of our optimized CoT examples in in-context learning (ICL), we provide a case study using the `Qwen-2.5-72B-Instruct` model on a representative problem from the MATH-500 dataset.

Figure 4 compares two responses: one directly generated by the `Qwen-2.5-72B-Instruct` model without any in-context examples (blue background), and the other generated under ICL using our optimized CoT example (pink background).

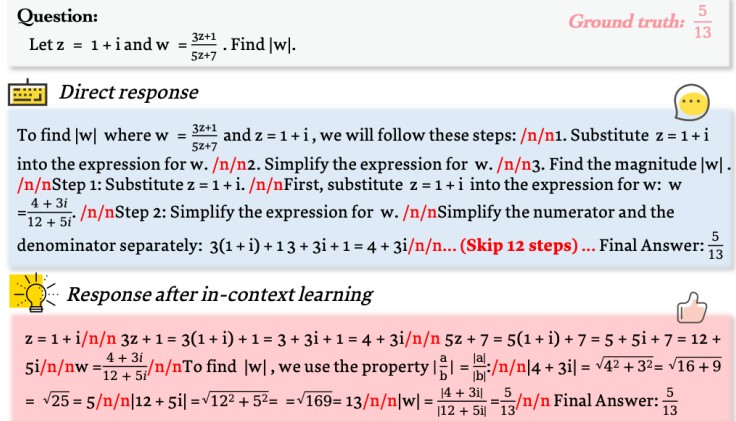

Figure 4: Case Study: Comparison of direct response from `Qwen-2.5-72B-Instruct` (blue background) and response under ICL with optimized CoT examples (pink background) on a MATH-500 problem. The optimized CoT enables more sufficient and necessary reasoning.

The direct response exhibits a lengthy reasoning process with several redundant or unnecessary steps and expressions. In contrast, the ICL-guided response is more concise and logically structured, reflecting a clearer and more efficient problem-solving strategy.

This comparison demonstrates how our optimized CoT exemplars help guide the model toward more focused and causally sufficient and necessary reasoning.

# F  Full Version of Related Work

## F.1  Reasoning Sufficiency Enhancement via CoT Optimization.

Recent efforts have focused on improving the reasoning capabilities of large language models (LLMs) through the development of Chain-of-Thought (CoT) reasoning and its variants. CoT [61] introduced intermediate reasoning steps to enable LLMs to perform structured, multi-step reasoning. This foundational idea has since evolved into more sophisticated frameworks such as Tree-of-Thought (ToT)[76] and Graph-of-Thought (GoT)[6], which organize reasoning structures into tree and graph

forms, respectively. Yue et al. [78] introduce DOTS, a method enabling dynamic reasoning trajectory planning via optimal reasoning strategy search, resulting in more adaptive and efficient problem-solving. Jin et al. [28] introduce a CoT framework using graph structures for iterative reasoning in LLMs and builds the GRBench dataset for graph-based reasoning evaluation. Yang et al. [69] explores how CoT prompts affect LLM mechanisms, enhancing knowledge retrieval by activating more neurons.

Several recent innovations further expand the reasoning capacity of LLMs. LLaVA-CoT [65] combines CoT with a multimodal visual-language model to enhance reasoning in vision-language tasks. Meta-CoT [64] formulates reasoning as a latent variable process, improving flexibility and generalization. Markov Chain of Thought [73] introduces Markovian transitions across reasoning steps by clearing the context KV cache to extend reasoning depth. Additionally, Puerto et al. [47] propose a self-correction mechanism using multiple reasoning paths, significantly improving performance on knowledge-intensive benchmarks. Hüyük et al. [26] propose fine-tuning strategies using counterfactual feedback to enhance LLMs' causal reasoning capabilities. Ma et al. [38] demonstrate that explicit thinking processes are not always necessary and propose a simplified "NoThinking" method achieving competitive reasoning performance with reduced computational costs. Diao et al. [18] present Active-Prompt, a method that enhances LLM reasoning by selectively annotating task-specific prompts for automatic adaptation. ECHO [31] unifies diverse reasoning paths to improve the consistency and accuracy of LLM reasoning.

Additional recent studies have explored novel dimensions in CoT reasoning. Ding et al. [19] propose "Break the Chain" strategies, integrating heuristic shortcuts to streamline CoT reasoning, significantly enhancing efficiency. Stolfo et al. [51] identify entropy and token frequency neurons, elucidating internal mechanisms by which LLMs manage uncertainty and confidence. Ali et al. [2] mitigate copy biases in in-context learning through targeted neuron pruning, improving generalization. De Sabbata et al. [15] use rational metareasoning to selectively invoke intermediate steps, reducing inference cost while preserving accuracy. Turpin et al. [58] question CoT's faithfulness by systematically evaluating model-generated rationales. Jin et al. [30] demonstrate that artificially lengthening reasoning can superficially boost performance, highlighting potential pitfalls in CoT evaluation. Simhi et al. [49] reveal that LLMs can exhibit high-certainty hallucinations, producing incorrect answers with strong confidence, thereby challenging the reliability of CoT-based outputs.

Despite their benefits, these structured reasoning methods often introduce excessive token length, which becomes problematic in cost-sensitive or latency-constrained scenarios [62]. Moreover, models frequently fail to assess task complexity, leading to over-reasoning on simple problems—an issue known as overthinking [12, 7, 36]. Our proposed approach addresses these limitations by simultaneously ensuring reasoning efficiency and maintaining high accuracy.

## F.2 Reasoning CoT Redundancy.

Recent studies on CoT redundancy have sought to mitigate redundancy in CoT reasoning. Token-budget-aware methods, such as Han et al. [22], dynamically allocate reasoning budgets based on task complexity. C3oT [32] leverages GPT-4 as a compressor to retain only essential reasoning content. CCoT [9] and COCONUT [23] adopt continuous representations to encode reasoning traces more compactly, while CoT-Valve [39] introduces variable-length CoTs. Training-free approaches, including Kimi K1.5 [54] and O1-Pruner [36], use prompt ensembles or reinforcement learning to discard unnecessary reasoning steps.

Recent efforts also investigate finer-grained control over CoT reasoning. Wu et al. [63] introduces the Thinking Intervention paradigm, allowing targeted interventions on reasoning tokens. Yang et al. [74] highlight the impact of excessive CoT length on model performance, proposing adaptive token scaling strategies. Liu et al. [35] demonstrate external thoughts from smaller models can effectively streamline reasoning in larger models, reducing redundancy.

However, most of these methods focus primarily on sequence length reduction or representation compression, without explicitly addressing causal logical redundancy. In contrast, our approach identifies and preserves only those reasoning steps that are both causally sufficient and necessary, grounded in a formal intervention-based framework. This results in CoT traces that are not only more compact but also more causally meaningful.

### F.3 PNS Theory in CoT Reasoning.

Our work is grounded in the causal inference framework proposed by Pearl [46], which defines the Probability of Sufficiency (PS) and the Probability of Necessity (PN) to quantify whether a cause is sufficient or necessary for a given effect. For instance, in image classification, PS measures how likely adding a feature (e.g., "pointy ears") leads to a positive label (e.g., "cat"), while PN measures how likely removing that feature would change the outcome.

While Hüyük et al. [26] introduced PN and PS as evaluation metrics to enhance model-level causal reasoning via counterfactual fine-tuning, our approach advances this direction by applying these metrics to the internal reasoning traces of Chain-of-Thought (CoT) prompting. Specifically, we formalize and operationalize the Probability of Necessary and Sufficient Causes (PNS) for individual reasoning steps, enabling a structured intervention-based analysis that identifies steps that are both causally essential and logically minimal.

Unlike prior work that emphasizes model-wide causal consistency, our framework targets step-level causal minimality within CoT, yielding concise, interpretable, and causally grounded reasoning sequences. This design allows us to optimize CoTs not merely for brevity or accuracy, but for causal soundness. Crucially, our method is model-agnostic and applies to any LLM capable of generating CoT outputs, representing a significant advancement over heuristic or model-specific compression strategies.

Our approach is essentially a causal analysis method for multi-step systems. In fact, it is not only applicable to Chain-of-Thought (CoT) reasoning, but can also inspire causal thinking in computational experiments [68, 77] and other complex systems.

## G  Additional PNS Comparison Results

This appendix provides comprehensive visualizations of average PNS values before and after applying our optimization algorithm across multiple models and datasets. Each figure presents results over 30 sampled problems, highlighting the increased causal necessity of the retained reasoning steps after optimization. See Figures 5 to 8 for detailed comparisons.

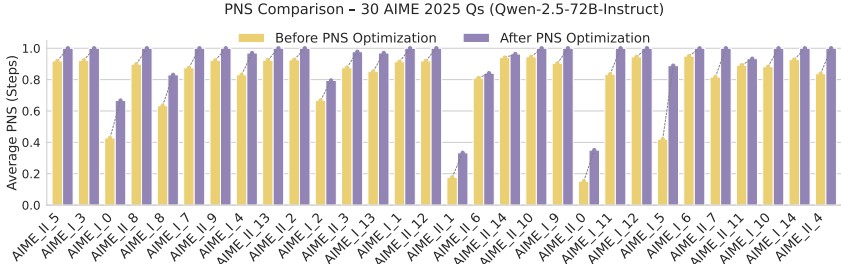

Figure 5: Average PNS comparison for `Qwen-2.5-72B-Instruct` on the AIME dataset.

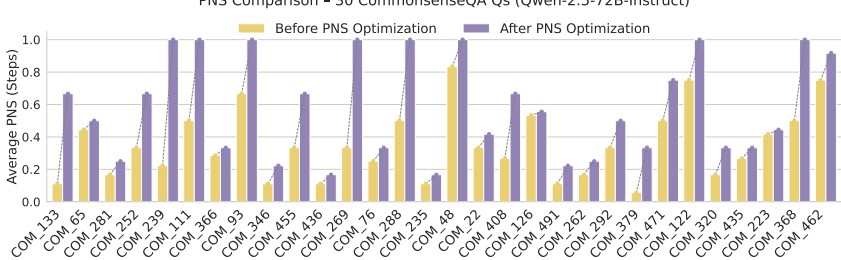

Figure 6: Average PNS comparison for `Qwen-2.5-72B-Instruct` on the CommonsenseQA dataset.

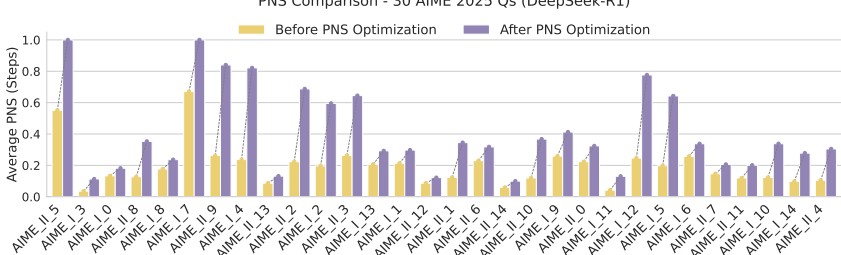

Figure 7: Average PNS comparison for `DeepSeek-R1` on the AIME dataset.

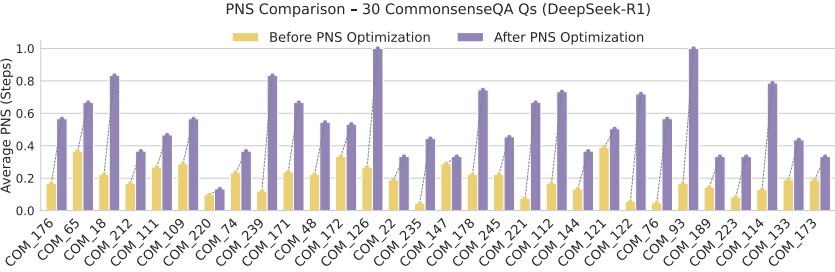

Figure 8: Average PNS comparison for `DeepSeek-R1` on the CommonsenseQA dataset.

# H    Computational Complexity Analysis

Let us denote the following quantities:

- $n$: the number of reasoning steps in the chain-of-thought (CoT);
- $k$: the number of rollouts per step;
- $l_{\text{step}}$: the average number of tokens generated per step;
- $l_{\text{out}}$: the number of tokens in each verification output (assumed constant);
- $t_{\text{token}}$: the time required to generate a single token (assumed constant).

We analyze the computational cost of our PNS-based intervention method by separately evaluating the *rollout* and *evaluation* stages.

## H.1    Rollout Time Complexity

At each reasoning step $i \in \{1, \ldots, n\}$, the model performs $k$ rollouts, where each rollout generates the remaining $(n - i)$ steps. Each step contains on average $l_{\text{step}}$ tokens. Thus, the total rollout time is given by:

$$T_{\text{rollout}} = \sum_{i=1}^{n} k \cdot (n - i) \cdot l_{\text{step}} \cdot t_{\text{token}} = O(k \cdot l_{\text{step}} \cdot t_{\text{token}} \cdot n^2) \tag{6}$$

## H.2    Evaluation Time Complexity

Each rollout must also be evaluated to determine its validity. This involves generating both the full continuation of the chain and a final answer output of length $l_{\text{out}}$. The total evaluation time is therefore:

$$T_{\text{eval}} = \sum_{i=1}^{n} k \cdot [(n - i) \cdot l_{\text{step}} + l_{\text{out}}] \cdot t_{\text{token}} = O(k \cdot t_{\text{token}} \cdot (l_{\text{step}} \cdot n^2 + l_{\text{out}} \cdot n)) \tag{7}$$

Since typically $l_{\text{out}} \ll l_{\text{step}} \cdot n$, the second term is asymptotically negligible, yielding:

$$T_{\text{eval}} = O(k \cdot l_{\text{step}} \cdot t_{\text{token}} \cdot n^2) \tag{8}$$

### H.3 Total Complexity

Combining the rollout and evaluation phases, the total computational cost becomes:

$$T_{\text{total}} = T_{\text{rollout}} + T_{\text{eval}} = O(k \cdot l_{\text{step}} \cdot t_{\text{token}} \cdot n^2) \tag{9}$$

Assuming that both $l_{\text{step}}$ and $t_{\text{token}}$ are constants (or near-constants) in practice, the overall time complexity simplifies to:

$$T_{\text{PNS}} = O(k \cdot n^2) \tag{10}$$

This quadratic dependence on $n$ highlights the computational cost of deeper reasoning chains, while the linear dependence on $k$ reflects the tradeoff between rollout breadth and computation time.

## I  Qualitative Analysis of Whether the Final Reasoning Is Sufficient and Necessary

We conducted a human evaluation of 50 chain-of-thought (CoT) samples generated by a model fine-tuned on data curated with our PNS-based algorithm. Five mathematics experts independently judged each sample under three labels:

- **S&N**: reasoning is both sufficient and necessary to support the final answer;
- **SbU**: reasoning is sufficient but contains unnecessary (redundant) steps;
- **NbI**: reasoning is insufficient (missing critical steps).

Table 5: Human evaluation of reasoning quality. "Fully Sufficient" = S&N + SbU. "Redundant" flags any chain containing redundant steps and may co-occur with NbI.

| Dataset | # Samples | Fully Sufficient | Redundant | S&N | SbU | NbI |
|---|---|---|---|---|---|---|
| GSM8K | 20 | 19 | 3 | 17 | 2 | 1 |
| Commonsense QA | 15 | 15 | 1 | 14 | 1 | 0 |
| MATH500 | 15 | 13 | 4 | 11 | 2 | 2 |
| **Total** | 50 | 47 (94.0%) | 8 (16.0%) | 42 (84.0%) | 5 (10.0%) | 3 (6.0%) |

Most outputs are logically sound under the sufficiency/necessity criterion: 84% of CoTs are both sufficient and necessary, and only 6% (3/50) are insufficient. MATH500 exhibits a higher incidence of redundancy and incompleteness, consistent with the greater difficulty of mathematical reasoning in this benchmark.

## J  Validator Accuracy and Robustness

To assess the reliability of the PNS validator $V$, we evaluate its performance across different LLMs and rollout sizes $k$. Specifically, we report the *mean absolute error* (MAE) between the estimated PNS values and ground-truth labels, where ground-truth PNS is defined as 1 for reasoning chains judged by experts to be both necessary and sufficient.

**Metric.** The evaluation computes the mean absolute difference $|\widehat{\text{PNS}} - \text{PNS}|$ across annotated samples. Lower values indicate better alignment with ground truth.

We compare validator accuracy across rollout sizes $k$ and model strengths. Increasing $k$ and using stronger LLMs both reduce MAE, yielding more stable and accurate PNS estimates. GPT-4o achieves the lowest error across all settings, underscoring that validator quality and model strength are key to reliable counterfactual analysis.

Table 6: Validator MAE Across LLMs and Rollout Sizes. Lower is better.

| Validator | $k=1$ | $k=3$ | $k=5$ | $k=10$ |
|---|---|---|---|---|
| Qwen-72B | 0.315 | 0.187 | 0.142 | 0.116 |
| Qwen-7B | 0.411 | 0.395 | 0.315 | 0.293 |
| GPT-4o | 0.137 | 0.114 | 0.090 | 0.050 |

# K    Supplementary Experimental Results

To further assess the robustness and generality of our proposed method, we compare it with three representative reasoning baselines: **SPIRIT**, **ReAct**, and **Tree-of-Thoughts (ToT)**. These methods represent distinct paradigms for enhancing large language model reasoning:

- **SPIRIT** [13] uses perplexity to identify key reasoning steps and prune redundant tokens.
- **ReAct** [75] integrates reasoning and acting by alternating between *thought* and *action* steps, improving interpretability and interactive decision-making.
- **Tree-of-Thoughts (ToT)** [76] introduces a tree-structured reasoning process, maintaining multiple reasoning trajectories to perform deliberate exploration and self-evaluation. We implement this process using a simple prompt-based approach.

Table 7 summarizes the results across three benchmarks—*CommonsenseQA*, *GSM-8K*, and *MATH-500*. Each metric is reported as **Tokens↓/ Steps↓/ Acc.↑**, where fewer tokens and steps indicate greater reasoning efficiency.

Table 7: Comparison of SPIRIT, ReAct, ToT, and Ours-ICL across models and datasets. Each metric is reported as **Tokens↓/ Steps↓ / Acc.↑**.

| Model | Method | CommonsenseQA | | | GSM-8k | | | MATH-500 | | |
|---|---|---|---|---|---|---|---|---|---|---|
| | | Tokens↓ | Steps↓ | Acc.↑ | Tokens↓ | Steps↓ | Acc.↑ | Tokens↓ | Steps↓ | Acc.↑ |
| DeepSeek-V3 | SPIRIT | 142.3 | 4.4 | **83.9** | 73.1 | **2.0** | 95.5 | 250.5 | 7.7 | 89.6 |
| | ReAct | 181.2 | 5.4 | 83.8 | 179.5 | 6.9 | 92.3 | 412.3 | 19.0 | 91.2 |
| | ToT | 271.8 | 8.2 | 80.1 | 198.9 | 7.7 | 91.7 | 349.1 | 13.9 | 72.8 |
| | Ours-ICL | **44.7** | **2.7** | 83.6 | **52.2** | 4.3 | **99.9** | **136.7** | **6.4** | **96.2** |
| Qwen-2.5-72B-Instruct | SPIRIT | 214.2 | 7.6 | 76.3 | 98.3 | **4.0** | 94.2 | 239.3 | 10.3 | **81.4** |
| | ReAct | 175.0 | 4.1 | **83.7** | 171.7 | 4.1 | 92.5 | 241.6 | **7.9** | 77.6 |
| | ToT | 269.2 | 7.9 | 80.0 | 273.5 | 10.4 | 75.1 | 345.1 | 14.1 | 70.6 |
| | Ours-ICL | **81.6** | **3.4** | 83.0 | **65.3** | 5.3 | **99.5** | **196.9** | 8.9 | 81.2 |
| Qwen-2.5-7B-Instruct | SPIRIT | 188.6 | 6.5 | 59.3 | 113.7 | 5.1 | 86.7 | 238.8 | 9.4 | **72.9** |
| | ReAct | 192.9 | 5.2 | **79.4** | 181.6 | 7.2 | 84.5 | 263.0 | 10.5 | 70.8 |
| | ToT | 335.4 | 10.1 | 78.0 | 212.3 | 8.9 | 70.5 | 302.6 | 12.3 | 50.0 |
| | Ours-ICL | **99.1** | **3.8** | 77.6 | **83.4** | **4.8** | **94.1** | **174.7** | **7.7** | 72.6 |
| LLaMA-3.1-8B-Instruct | SPIRIT | 170.1 | 7.4 | 63.2 | 179.7 | **7.8** | 80.9 | 1218.1 | 67.3 | 42.8 |
| | ReAct | 382.1 | 17.9 | **79.0** | 521.5 | 20.6 | 82.5 | 1176.9 | 53.6 | 46.0 |
| | ToT | 559.9 | 19.4 | 75.3 | 711.7 | 27.1 | 69.0 | 996.7 | 45.8 | 30.2 |
| | Ours-ICL | **120.3** | **7.1** | 72.1 | **128.3** | 8.1 | **93.1** | **365.9** | **24.2** | **54.8** |

Across all models and datasets, our proposed **Ours-ICL** consistently achieves a superior balance between reasoning accuracy and efficiency. While ReAct and ToT obtain competitive accuracy on commonsense reasoning tasks, they dramatically increase reasoning cost in both tokens and steps. SPIRIT improves efficiency via perplexity-based step selection but can underperform on complex mathematical reasoning. In contrast, **Ours-ICL** yields the best overall trade-off, reducing reasoning cost by up to 60–80% while maintaining or improving accuracy, especially on *GSM-8K* and *MATH-500*.

