# OpenReview forum: "Causal Sufficiency and Necessity Improves Chain-of-Thought Reasoning"
_NeurIPS.cc/2025/Conference — NeurIPS 2025 poster_

### Official Review · Reviewer_Br1J · 2025-06-02

**Clarity:** 3
**Significance:** 2
**Originality:** 3
**Rating:** 5
**Confidence:** 3

**Summary:**

The manuscript proposes to refine Chain of Thought (CoT) reasoning by scoring explanations by causality-inspired metrics designed to track the probabilities of necessity and sufficiency, as originally described by Pearl. This should encourage CoTs that are as short and informative as possible, ideally providing all and only the steps required to answer the relevant query. Experiments demonstrate the effectiveness of the method in improving reasoning in mathematical and common sense tasks using a range of LLMs.

**Questions:**

$\textbf{Major comments}$

-The causal allusion is suggestive, but it's not clear to me how it really applies if there is no counterfactual reasoning or structural dependence between variables (unless this is secretly tucked into the mysterious validator $V$ somehow?) Is there an abduction step somewhere that I'm missing? Put another way: would anything about this method be different if we replaced Pearl's probabilities of causation with standard conditional probabilities such as $p(a \mid s, q)$ and $p(a' \mid s', q)$? Perhaps this is what the causal vs. noncausal experiment in Table 3 is meant to summarize, but I was not clear on what exactly differentiates these two methods. It's very possible I've missed something here, and I would gladly revise my score upward if the authors could clarify the uniquely causal component of their procedure and/or explain how $V$ operationalizes PNS.

-I expected to see some discussion of the complexity overheads associated with this approach (if not a theoretical analysis, at least a run time comparison). I assume that optimizing for PNS incurs some extra compute? This is potentially important given that CoT is already a very computationally intensive procedure.

$\textbf{Minor comments}$

-How many rollouts per intervention are typically needed to get reliable PNS estimates in practice?

-In Eq. 4, I assume that $q$ should be a conditioning event on the RHS?

-There is a whole literature on necessity and sufficiency for XAI that may be relevant here (see links below).

-https://proceedings.mlr.press/v177/beckers22a.html

-https://link.springer.com/article/10.1007/s11023-022-09598-7

-https://link.springer.com/article/10.1007/s10849-022-09377-8

-https://dl.acm.org/doi/10.1145/3448016.3458455

-https://dl.acm.org/doi/10.1145/3351095.3372850

**Ethical Concerns:**

["NO or VERY MINOR ethics concerns only"]

**Final Justification:**

The authors did an excellent job responding to my comments, as well as those of the other reviewers. I had missed some important aspects of the contribution in my original review that were fully clarified during the rebuttal stage. The extra experiments were also quite helpful. In light of these updates, I have decided to revise my score upward.

**Limitations:**

Yes

**Quality:**

3

**Strengths And Weaknesses:**

$\textbf{Strengths}$: the paper is clear and easy to follow. The central idea is simple and compelling. It stands to reason that users would generally prefer CoTs that maximize the probability of necessity and sufficiency (PNS) over those that do not, all else being equal. The empirical results are strong.

$\textbf{Weaknesses}$: while the inspiration is clear enough, the execution appears rather heuristic, as far as I can tell. The "validation model" $V$ is doing all the heavy lifting here – we literally just equate PNS with the complement of its average – but I see no reason to believe that $V$ is actually measuring PNS, even indirectly (see comments below).

---

> ### Author Rebuttal · Authors · 2025-07-30
>
> **1.[Q1] How does your method implement causal reasoning—if there is no explicit structural dependence or counterfactual modeling? Would the results differ if standard conditional probabilities were used instead?**
>
> **(1) Why Our Method Is Counterfactual**
>
> Counterfactual reasoning follows the “**abduction–intervention–prediction**” paradigm. In our case:
>
> - **Abduction** is implicitly simulated by fixing the upstream context $(s_{<t}, q)$ for all rollouts.
> - **Intervention** is performed by replacing a single reasoning step $s_t$ with a semantically disjoint alternative $\bar{s}_t$.
> - **Prediction** is executed by generating a full downstream reasoning chain from the modified input and evaluating the outcome.
>
> While we do not build a causal graph explicitly, this process **mimics counterfactual inference** by isolating a cause ($s_t$) and observing the impact of its intervention ($\bar{s}_t$) on the final answer. This is distinct from simply computing conditional probabilities like $p(a \mid s, q)$ or $p(a’ \mid \bar{s}, q)$, which lack an explicit notion of intervention or structural manipulation. In our setting, we **actively intervene** on the step and generate downstream consequences.
>
> **(2) How PNS Is Computed Operationally**
>
> While PNS is theoretically defined in terms of counterfactual probabilities, it is **not directly computable**. In our paper, we derive an empirical approximation (see paper Equation 10):
>
> $$
> \mathrm{PNS}(\mathbf{S}, \overline{\mathbf{s}}\_t, \mathbf{q}) \approx 1 - \frac{1}{k} \sum_{i=1}^{k} V(\overline{\mathbf{S}}^{(i)})
> $$
>
> Here’s how we compute it in practice:
>
> 1. We perform a **counterfactual intervention** on a single step $s_t$ by replacing it with a semantically disjoint alternative $\bar{s}_t$ (i.e., not a trivial paraphrase).
> 2. From the alternative step, we use a large language model to generate a full reasoning chain $\bar{S}_{>t}$.
> 3. We apply a validator $V$ to check whether the counterfactual reasoning still leads to the correct final answer.
>
> The validator $V$ is **not a static function, but rather an operation** that executes the new reasoning path and checks its correctness.
>
> We repeat this process for $k$ different $\bar{s}_t^{(i)}$, and estimate the **counterfactual success rate**:
>
> $$
> \frac{1}{k} \sum_{i=1}^{k} V(\overline{\mathbf{S}}^{(i)})
> $$
>
> This success rate quantifies **how easily the original step $s_t$ can be replaced without affecting the final outcome**:
>
> - A high success rate implies that $s_t$ is not necessary—it can be replaced with alternatives that still lead to the correct result.
> - A low success rate suggests that $s_t$ is causally necessary—its absence results in incorrect answers.
>
> Thus, the complement term 1 - $\text{success rate}$ serves as an empirical estimate of the PNS value for step $s_t$.
>
> **(3) Validator Accuracy and Robustness**
>
> To assess the reliability of the validator $V$, we conduct experiments varying both the underlying LLM and the number of rollouts $k$. The table below shows the **mean absolute error** between estimated PNS values and ground-truth, where ground-truth PNS is set to 1 based on a reasoning chain judged by experts to be both necessary and sufficient.
>
> **Table R1.** Validator Accuracy Across LLMs and Rollout Sizes
> >Values indicate the mean absolute error $| \widehat{\mathrm{PNS}} - \mathrm{PNS} |$ across evaluation samples. Lower is better.
>
> | **Validator** | **k = 1** | **k = 3** | **k = 5** | **k = 10** |
> | ------------- | --------- | --------- | --------- | ---------- |
> | Qwen-2.5-72B-Instruct  | 0.315     | 0.187     | 0.142     | 0.116      |
> | Qwen-2.5-7B-Instruct   | 0.411     |  0.395   |  0.315   |  0.293    |
> | GPT-4o        | 0.137     |   0.114  | 0.090     | 0.050      |
>
> - Increasing $k$ improves the stability and reliability of the PNS estimate.
> - Stronger LLMs used as validators yield more accurate evaluations.
> - The validator $V$ is crucial for enabling **counterfactual analysis** in practice.
>
> We will include this explanation in the revised manuscript.
>
> **2.[Q2] Analysis of the Number and Complexity of Tokens for Complete PNS Reasoning.**
>
> (1) We **added a new experiment** (see **Table R2**) showing token cost and rollout complexity across LLMs (e.g., Qwen2.5) and LRMs (e.g., DeepSeek R1) during PNS filtering.
>
> (2) We also compare average tokens per CoT before and after PNS-based fine-tuning (see **Table 3 in the main paper)**, demonstrating reduced inference cost and improved conciseness.
>
> - Across three different LLM families, we observe a consistent trend: while PNS filtering introduces a moderate one-time token cost during data preparation, it yields over 20% token savings during inference.
> - The accuracy of all models is either maintained or improved, confirming that the CoTs retained after causal filtering are both concise and effective.
> - This supports our central claim: PNS filtering reshapes the training distribution, enabling the model to generate more efficient and logically minimal reasoning paths—without requiring inference-time pruning.
>
> **Table R2**. Token usage and complexity during PNS filtering
> | **Model (Base)**            | **Dataset**       | **Rollouts** | **Avg. Tokens** | **Time Complexity** |
> |----------------------------|-------------------|--------------|------------------|---------------------|
> | Qwen-2.5-72B-Instruct | GSM8k             | 9.77         | 1,104.42         | $O(k · n²)$          |
> |                            | CommonsenseQA     | 24.84        | 2,945.61         | $O(k · n²)$           |
> |                            | MATH-500          | 21.66        | 2,315.26         | $O(k · n²)$          |
> |                            | AIME              | 61.81        | 12,145.87        | $O(k · n²)$         |
> | DeepSeek R1            | GSM8k             | 22.62        | 2,861.12         | $O(k · n²)$         |
> |                            | CommonsenseQA     | 30.81        | 4,501.99         | $O(k · n²)$         |
> |                            | MATH-500          | 46.16        | 9,855.70         | $O(k · n²)$         |
> |                            | AIME              | 660.90       | 371,430.57       | $O(k · n²)$          |
> > *Rollouts*: Average number of step-level interventions (regenerations) per CoT.
> *Avg. Tokens*: Average total number of tokens generated per example during PNS filtering.
>
> Overall, our results show that the proposed method **reduces reasoning redundancy**, trading a one-time overhead for sustained improvements in inference efficiency.
>
> **3.[Q3] How many rollouts per intervention are typically needed to get reliable PNS estimates in practice?**
>
> As shown in **Table R1(above)**, we conducted additional evaluations to assess how the number of rollouts  $k$  affects the stability of PNS estimates. Both theoretical considerations and empirical results indicate that increasing $k$ improves estimation reliability. In practice, we found that using $k = 3 \sim 5$ strikes a good balance between computational cost and accuracy. We will clarify this design choice in the final version.
>
> **4.[Q4] In Equation 4, should q be included as a conditioning variable on the right-hand side?**
>
> Thank you for pointing this out. Yes, all probability expressions in our method are implicitly conditioned on the question $q$ (e.g., Equation 4). We followed the standard convention of omitting global conditions like the input prompt for notational simplicity. In the revised manuscript, we will explicitly clarify that all probability terms are conditioned on $q$, unless otherwise specified.
>
> **5.[Q5] There is a whole literature on necessity and sufficiency for XAI that may be relevant here (see links below).**
>
> Thank you very much for the valuable references — these are all excellent and inspiring works. We will incorporate them into the **Related Work** section of the revised version.
> >  Causal Necessity and Sufficiency in XAI. Our work relates closely to prior studies that explain model behavior via necessity and sufficiency [1–5]. For instance, LENS [1] identifies necessary and sufficient conditions for outputs better than mainstream XAI methods; Darwiche et al. [2] use Decision-DNNF circuits to compute all sufficient reasons efficiently; Mothilal et al. [3] generate diverse counterfactuals using determinant point processes; Beckers [4] formalizes sufficiency-based explanations for action guidance and fairness; Galhotra et al. [5] introduce LEWIS, a probabilistic counterfactual explanation method.
> In contrast, our PNS evaluation targets LLM reasoning chains, using counterfactual rollouts to assess the causal necessity and sufficiency of CoT traces—offering a causal solution to the overthinking problem in LLMs.

---

> > ### Comment · Reviewer_Br1J · 2025-08-03
> > **Reply to authors**
> >
> > Many thanks to the authors for their detailed reply to my comments. I have also read the other reviews and the ensuing discussions, which I found quite helpful. I am satisfied that my primary concerns have been addressed and will revise my score upward to reflect this.

---

> > > ### Author Response · Authors · 2025-08-05
> > >
> > > We sincerely thank the reviewer for the thoughtful reconsideration and kind words!  We truly appreciate your updated evaluation and are grateful that our responses addressed your concerns.

---

### Official Review · Reviewer_iGwQ · 2025-06-29

**Clarity:** 2
**Significance:** 2
**Originality:** 3
**Rating:** 4
**Confidence:** 4

**Summary:**

This paper proposes to characterize CoT with causal probability of sufficiency and necessity. In a CoT, the intermediate steps are sufficient if they can support the conclusion, while necessary if they are indispensable for the soundness of the final answer. By optimizing the sufficiency and necessity of CoT (i.e., PNS optimization) with causal intervention on each intermediate step (actually, the authors only optimized the necessity), the authors reconstruct more concise CoT responses from training data, eliminating redundant steps without affecting the accuracy. Empirically, the PNS optimization method enhances causal sufficiency and necessity of CoTs. While the optimized CoTs contain fewer tokens and steps, they don’t compromise the performance and can sometimes even improve the performance on reasoning benchmarks in both in-context learning and finetuning scenarios.

**Questions:**

1. Can you discuss how PNS optimization is different from SPIRIT[1]? They look similar as both involve counterfactual intervention and scoring with an LLM.
2. In Equation 2, what is $\bar{S}$ in the RHS? It’s not defined in the LHS. Can I understand RHS as the average for any $\bar{S}$?
3. Is Equation 4 factorizable as $P(A_S = y) P(A_{S'} \neq y)$? That would be more intuitive for your Lemma 1.
4. What is semantic disjointness in Algorithm 1?
5. Where do the unoptimized CoT traces come from? Are they human annotated, or elicited by zero-shot CoT prompting? If there is no comparison against other CoT optimization methods, how can I trust the initial unoptimized CoT traces are not deliberately made overthinking?
6. What is the setting for measuring accuracy in Table 1? Is it like in-context learning in Table 2 or finetuning in Table 3?
7. Table 3: Does original refer to the model before SFT? Does noncausal refer to unoptimized SFT data?
8. Line 97: in Pearl → by Pearl

**Ethical Concerns:**

["NO or VERY MINOR ethics concerns only"]

**Final Justification:**

The authors have successfully addressed my concern on lack of baselines in their response. Regarding the claim of optimizing sufficiency, I assume it's overclaim given that the authors didn't directly respond to my concern in their response. I am generally comfortable with my rating and don't oppose the acceptance of this paper.

**Limitations:**

Yes.

**Paper Formatting Concerns:**

No formatting issues.

**Quality:**

3

**Strengths And Weaknesses:**

Strengths:

1. As CoTs grow longer and longer in recent LLMs, it is very important to optimize the length of CoTs to reduce finetuning and inference workloads. This paper provides an automatic solution for optimizing CoTs without human annotation.
2. Unlike previous methods that mostly rely on correlation-based metrics, this paper uses causal metrics to guide the optimization of CoT, which is intuitive and may inspire new methods in this field.
3. Experimental results show that the proposed PNS optimization can reduce the length of CoTs by 2-6x, without compromising performance in both in-context learning and finetuning scenarios. However, it’s unclear whether the proposed method is effective or CoT optimization is easy (see weakness 1).

Weaknesses:

1. The authors only compared PNS optimization against efficient CoT prompting methods. I feel efficient CoT prompting methods might be weak baselines here, since they are just slightly optimized by human. It’s not clear whether the proposed method is effective, or standard prompts are so redundant that any CoT optimization method can achieve good results. The authors need to consider other CoT optimization methods, such as SPIRIT[1], or a CoT optimization method derived from information gain[2]. [1, 2] should also be discussed in related works.
2. While the authors claim to optimize sufficiency and necessity, in reality they assumed sufficiency is fulfilled and only optimized necessity, as shown in Algorithm 1 and Lemma 1. In that sense, the “addition of missing steps” in the abstract, as well as the “bi-level optimization framework” in the contribution part, are overclaim.
3. The writing of this paper has much room to improve. The title may be more specific about what causal sufficiency and necessity improves. The authors may discuss the impact of optimizing CoTs and then transit to the challenges of sufficiency and necessity.

[1] Cui et al. Stepwise Perplexity-Guided Refinement for Efficient Chain-of-Thought Reasoning in Large Language Models. arXiv 2025.

[2] Ton and Taufiq et al. Understanding Chain-of-Thought in LLMs through Information Theory. arXiv 2024.

---

> ### Author Rebuttal · Authors · 2025-07-30
>
> We sincerely thank you for your constructive and insightful feedback. Our responses are as follows:
>
> **1. [Q1] Can you discuss how PNS optimization is different from SPIRIT [1]? They look similar as both involve counterfactual intervention and scoring with an LLM.**
>
> Thank you for raising this important point. We have conducted an additional experiment to directly compare our method with SPIRIT, focusing on its prompt-based ICL version. The results are shown in Table R1 below, and more baseline reproductions will be added in future work.
>
> Although both our method and SPIRIT [1] involve selecting reasoning steps, there are **fundamental differences** in their principles and goals:
>
> 1. **Causal vs. Heuristic**: Our method is grounded in **causal theory**, using Probability of Necessity and Sufficiency (PNS) to guide selection. In contrast, SPIRIT relies on **perplexity**, a fluency-based heuristic that reflects the LLM’s confidence but not its causal contribution.
> 2. **Counterfactual Intervention**: We perform actual interventions (i.e., step modification and rollout) to estimate how altering each step impacts the final outcome. SPIRIT does not perform such counterfactual evaluation.
> 3. **Optimization Objective**: Our goal is to **retain only those steps that are both necessary and sufficient**, ensuring causal minimality. SPIRIT, on the other hand, aims to **reduce token count while maintaining fluency**, which may inadvertently discard causally important steps.
>
> We have added both SPIRIT [1] and Ton et al. [2] to the Related Work section of the revised manuscript:
> > SPIRIT [1] uses perplexity to effectively identify key reasoning steps, achieving a balance between accuracy and efficiency in both few-shot CoT and fine-tuning settings, and also exhibits strong cross-model transferability. Ton et al. [2], based on information-theoretic principles, quantify the contribution of each step in the reasoning chain to the final correct answer using conditional mutual information, enabling the identification of reasoning failure patterns in large language models without requiring manual annotation of intermediate steps.
>
> **Table R1.** Comparison with SPIRIT.
> | **Model**                | **Method** | **CommonsenseQA (Tokens / Steps / Accuracy)** | **GSM-8k (Tokens / Steps / Accuracy)** | **MATH-500 (Tokens / Steps / Accuracy)** |
> | ------------------------- | -------- | ------------------------------------------- | ------------------------------------ | -------------------------------------- |
> | **DeepSeek-V3**           | SPIRIT   | 142.3/4.4/**83.9%**                         | 73.1/**2.0**/95.5%                   | 250.5/7.7/89.6%                        |
> |                           | Standard | 177.5/5.7/83.8%                             | 157.3/7.4/97.6%                      | 598.6 /26.7 /93.2%                     |
> |                           | Ours-ICL | **44.7**/ **2.7**/83.6%                     | **52.2** /4.3 /**99.9%**             | **136.7**/**6.4**/**96.2%**            |
> | **Qwen-2.5-72B-Instruct** | SPIRIT   | 214.2/7.6/76.3%                             | 98.3/**4.0**/94.2%                   | 239.3/10.3/81.4%                       |
> |                           | Standard | 109.1/3.7 /78.2%                            | 113.8/8.0 /93.6%                     | 281.8/9.2/**84.0%**                    |
> |                           | Ours-ICL | **81.6** /**3.4** /**83.0%**                | **65.3**/5.3 /**99.5%**              | **196.9** /**8.9**/81.2%               |
> | **Qwen-2.5-7B-Instruct**  | SPIRIT   | 188.6/6.5/59.3%                             | 113.7/5.1/86.7%                      | 238.8/9.4/**72.9%**                    |
> |                           | Standard | 209.8/7.2 /70.4%                            | 149.7/7.2/85.1%                      | 263.5 /9.7/71.0%                       |
> |                           | Ours-ICL | **99.1** /**3.8**/**77.6%**                 | **83.4**/**4.8**/**94.1%**        | **174.7**/**7.7**/72.6%                |
> | **LLaMA-3.1-8B-Instruct** | SPIRIT   |        170.1/7.4/63.2%     |      179.7/**7.8**/80.9%      |      1218.1/67.3/42.8%             |
> |                           | Standard | 169.3 /7.3 /**72.2%**                         | 182.8/7.9/79.2%                      | 741.3 /46.0 /47.6%                     |
> |                           | Ours-ICL | **120.3**/**7.1**/72.1%                           | **128.3**/8.1/**93.1%**                      | **365.9** /**24.2** /**54.8%**                     |
>
> Our additional experiments corroborate the main findings reported in the paper (see lines 212–222): **our method consistently achieves better accuracy–efficiency trade-offs across diverse models and datasets**. Compared to several baselines, it significantly reduces the number of steps and tokens without compromising accuracy—and often improves it.
>
> **2.[Q2]  In Equation 2, what is $\bar{S}$ in the RHS? It’s not defined in the LHS. Can I understand RHS as the average for any $\bar{S}$?**
>
> Thank you for pointing this out. In Equation 2,  $\bar{S}$ denotes a **reasoning chain that fails to produce the correct answer**. The expression on the right-hand side should not be interpreted as an average over all possible , but rather as being restricted to the context where $A \ne y$.
>
> **3. [Q3] Is Equation 4 factorizable as $P(A_S = y) \cdot P(A_{S'} \ne y)$? That would be more intuitive for your Lemma 1.**
>
> Thank you for the question. This factorization is mathematically invalid due to a causal dependency between the events $A_S = y$ and $A_{S'} \ne y$.
>
> Equation (4) defines a joint probability:
>
> $$
> PNS(S, s_t, q) = P(A_S = y, A_{S'} \ne y)
> $$
>
> By the chain rule of probability:
>
> $$
> P(A_S = y, A_{S'} \ne y) = P(A_S = y) \cdot P(A_{S'} \ne y \mid A_S = y)
> $$
>
> Since $S$ and $S'$ differ by only one step, their outputs are highly correlated. Therefore, the conditional probability **cannot be simplified** to a marginal:
>
> $$
> P(A_{S'} \ne y \mid A_S = y) \ne P(A_{S'} \ne y)
> $$
>
> We will include this derivation and explanation in the final version of the paper for clarity.
>
> **4. [Q4] What is semantic disjointness in Algorithm 1?**
>
> In Algorithm 1, semantic disjointness refers to the requirement that the perturbed step and its subsequent steps are substantially different in meaning and reasoning logic from the original step. This constraint avoids trivial rewordings and ensures that the counterfactual chain represents a truly distinct reasoning path. We will clarify this in the algorithm description.
>
> **5. [Q5] Where do the unoptimized CoT traces come from? Are they human annotated, or elicited by zero-shot CoT prompting? If there is no comparison against other CoT optimization methods, how can I trust the initial unoptimized CoT traces are not deliberately made overthinking?**
>
> Thank you for the question. The unoptimized CoTs used in our experiments are generated via zero-shot or few-shot prompting from strong language models (e.g., Qwen-2.5-72B-Instruct). These are produced using the same system prompt as in our Ours-ICL setting (see Appendix C), and **no human editing or annotation** is applied. To avoid bias, we use standard open-ended prompts and do not manually craft overthinking chains.
>
> We have already included **multiple prompt-based baselines** for comparison (see **Table 2** in the paper main text and prompt templates in **Appendix C**). In this revision, we also added SPIRIT as a new baseline in the expanded **Table R1** (which extends Table 2 of the main paper).
>
> **6. [Q6] What is the setting for measuring accuracy in Table 1? Is it like in-context learning in Table 2 or fine-tuning in Table 3?**
>
> Accuracy is consistently computed as follows, and the setting corresponds to in-context learning as in Table 2 or fine-tuning as in Table 3, depending on the experiment.
> $
> \text{Acc} = \frac{\text{Number of correct CoTs}}{\text{Total number of CoTs in the dataset}}.
> $
>
> The accuracy improvements in Table 1 arise from repeated applications of Algorithm 1 across the dataset, where each step in the CoT undergoes multiple rollouts. This significantly increases the likelihood of generating a correct answer. CoTs with PS < 1 may be ignored in a single run, but will be revisited in subsequent rollouts. Thus, PS enhancement is implicitly achieved via repeated generation and validation within each CoT execution unit.
>
> **7. [Q7] Table 3: Does “Original” refer to the model before SFT? Does “Noncausal” refer to unoptimized SFT data?**
>
> Yes, exactly:
>
> - “**Original**” refers to the base model before any supervised fine-tuning (SFT).
> - “**Noncausal**” refers to the model fine-tuned on unfiltered, unoptimized CoTs.
> - “**Causal**” refers to the model fine-tuned on CoTs filtered and optimized via our PNS method.
>
> **8.[Q8] Line 97: “in Pearl” → “by Pearl”**
>
> Thank you for catching this! We will correct “in Pearl” to “by Pearl” in the revised version.

---

> > ### Author Response · Authors · 2025-08-04
> > **We would like to supplement more experiment results to fully address your concerns on the evaluation completeness.**
> >
> > We would like to **supplement more experiment results** to fully address your concerns on the evaluation completeness.
> >
> > **Table R2.** ReAct Results.
> >
> > | Model                     | Method   | CommonsenseQA (Tokens / Steps / Accuracy) | GSM-8k (Tokens / Steps / Accuracy) | MATH-500 (Tokens / Steps / Accuracy) |
> > | ------------------------- | -------- | ----------------------------------------- | ---------------------------------- | ------------------------------------ |
> > | **DeepSeek-V3**           | ReAct[1] | 181.2 / 5.4 / 83.8%                       | 179.5 / 6.9 / 92.3%                | 412.3 / 19.0 / 91.2%                 |
> > | **Qwen-2.5-72B-Instruct** | ReAct[1] | 175.0 / 4.1 / 83.7%                       | 171.7 / 4.1 / 92.5%                | 241.6 / 7.9 / 77.6%                  |
> > | **Qwen-2.5-7B-Instruct**  | ReAct[1] | 192.9 / 5.2 / 79.4%                       | 181.6 / 7.2 / 84.5%                | 263.0 / 10.5 / 70.8%                 |
> > | **LLaMA-3.1-8B-Instruct** | ReAct[1] | 382.1 / 17.9 / 79.0%                      | 521.5 / 20.6 / 82.5%               | 1176.9 / 53.6 / 46.0%                |
> >
> > **Table R3.** ToT Results.
> >
> >
> > | Model                     | Method  | CommonsenseQA (Tokens / Steps / Accuracy) | GSM-8k (Tokens / Steps / Accuracy) | MATH-500 (Tokens / Steps / Accuracy) |
> > | ------------------------- | ------- | ----------------------------------------- | ---------------------------------- | ------------------------------------ |
> > | **DeepSeek-V3**           | ToT\[2] | 271.8 / 8.2 / 80.1%                       | 198.9 / 7.7 / 91.7%                | 349.1 / 13.9 / 72.8%                 |
> > | **Qwen-2.5-72B-Instruct** | ToT\[2] | 269.2 / 7.9 / 80.0%                       | 273.5 / 10.4 / 75.1%               | 345.1 / 14.1 / 70.6%                 |
> > | **Qwen-2.5-7B-Instruct**  | ToT\[2] | 335.4 / 10.1 / 78.0%                      | 212.3 / 8.9 / 70.5%                | 302.6 / 12.3 / 50.0%                 |
> > | **LLaMA-3.1-8B-Instruct** | ToT\[2] | 559.9 / 19.4 / 75.3%                      | 711.7 / 27.1 / 69.0%               | 996.7 / 45.8 / 30.2%                 |
> >
> > The supplementary experimental results, as **shown in Table R2 and Table R3**, indicate that the two newly added methods perform well in terms of accuracy on CommonsenseQA, but underperform on math-related datasets. Moreover, these methods significantly increase the number of reasoning Tokens and Steps across all datasets, resulting in noticeably reduced efficiency. In contrast, when compared against all seven baseline methods, our proposed **ours-ICL consistently achieves higher accuracy and lower reasoning cost across multiple datasets,** striking a better balance between accuracy and efficiency.
> >
> > [1] Yao S, Zhao J, Yu D, et al. React: Synergizing reasoning and acting in language models[C]//International Conference on Learning Representations (ICLR). 2023.
> >
> > [2] Yao S, Yu D, Zhao J, et al. Tree of thoughts: Deliberate problem solving with large language models[J]. Advances in neural information processing systems, 2023, 36: 11809-11822.

---

> > > ### Comment · Reviewer_iGwQ · 2025-08-04
> > >
> > > Thanks the authors for their proper response. The authors have addressed my concern on lack of baselines. However, I didn't find the answer to my weakness 2 in the response. Can the authors explain where sufficiency is optimized in their proposed method? If not, please remove that claim from the paper.

---

> > > > ### Author Response · Authors · 2025-08-06
> > > > **A more detailed response on "where sufficiency is optimized in their proposed method?"**
> > > >
> > > > Thank you for your detailed feedback and we are delighted that we have cleared your previous concerns!
> > > >
> > > > For then remaining question that *“Can the authors explain where sufficiency is optimized in their proposed method?”*, we would like to supplement a more detailed explanation to make our reasoning more well-motivated.
> > > >
> > > > Note that our method is based on the definition of PNS in terms of counterfactual language:
> > > >
> > > > $$
> > > > \mathrm{PNS}(\mathbf{S}, \overline{\mathbf{s}}\_t,\mathbf{q}) := P\left(\mathbf{A\_{S}=\mathbf{y}, A\_{{S'}} \neq y}\right).
> > > > $$
> > > >
> > > > **The original PNS is intractable because it defines on the joint counterfactual probabilities.**
> > > >
> > > > Thus, we provided the **empirical approximation** as below (Eq. 5 in our paper), this approximation is based on an assumption of $P(\mathbf{A=y}\mid \operatorname{do}(\mathbf{S}),\mathbf{q})=1$ (fully sufficiency), which is equavalent with when PS of the CoT chain is equals to 1 (Theorem 1 in Appendix):
> > > > $$
> > > > \mathrm{PNS}(s_t) \approx 1 - \frac{1}{k} \sum_{i=1}^k V(\overline{\mathbf{S}}^{(i)}).
> > > > $$
> > > >
> > > > Based on the tractable emperical approximation above, to evaluate PNS, **we should firstly have to make sure the chain satisfy the fully sufficiency assumption** of $P(\mathbf{A=y}\mid \operatorname{do}(\mathbf{S}),\mathbf{q})=1$. Thus, in this framework, **our method improves PS before PN estimation** and make it as a bi-level optimization process via the following process:
> > > >
> > > > ## 1. Iterative Resampling to Improve PS
> > > >
> > > > **If the original reasoning chain  $S$  is not sufficient, we re-run Algorithm 1 to sample a new chain  $S'$  under the same question context $(q, s_{<t})$**. Each re-run explores a different reasoning trajectory, increasing the chance of obtaining a causally sufficient chain.
> > > > This process is repeated multiple times, and with each re-run, the probability of having at least one sufficient reasoning chain increases:
> > > > $$
> > > > P_{\text{hit}}(n) = 1 - (1 - p_{\text{suf}})^n,
> > > > $$
> > > > where  $p_{\text{suf}}$  is the sufficiency probability of a single rollout. Even without aiming for PS = 1 , multiple re-runs steadily raise the overall sufficiency level in the candidate set, ensuring that the subsequent PN estimation is not dominated by insufficient reasoning.
> > > >
> > > > ## 2. PN Computation After PS Enhancement
> > > >
> > > > **Once a set of reasoning chains with improved sufficiency has been obtained, we proceed to compute PN** via counterfactual interventions (Algorithm 1):
> > > > - Replace  $s_t$ with multiple semantically disjoint alternatives  $\bar{s}_t^{(i)}$ ,
> > > > - Generate downstream reasoning  $\overline{\mathbf{S}}^{(i)}$ ,
> > > > - Validate each with $V(\cdot)$  to estimate the counterfactual success rate.
> > > > Because PS has already been raised through iterative resampling, the PN estimate is less affected by incidental errors unrelated to the necessity of  $s_t$.
> > > >
> > > > ## 3. Why This Improves PNS Reliability
> > > >
> > > > By raising PS then repeating execution of Algorithm 1 (PN computation), our PNS estimation benefits in two ways:
> > > > - **Reduced bias** – PN is calculated on reasoning chains that are already more likely to be correct, preventing underestimation due to irrelevant failures.
> > > > - **Better robustness** – Even without achieving perfect sufficiency, the improved PS ensures a more accurate PNS calculation.
> > > >
> > > > In summary, **PN is explicitly computed via interventions, while PS is implicitly optimized through multiple re-runs of Algorithm 1** (We will update this re-run operation to Algorithm 1 to make it more clear). This two-stage process ensures that PNS estimation reflects both necessity and sufficiency in a balanced and computationally practical manner.
> > > >
> > > > ***
> > > >
> > > > Please let us know if we have resolved your concerns – thank you!

---

> ### Author Response · Authors · 2025-08-05
> **Clarifying the Optimization of Sufficiency in Our Method**
>
> We sincerely thank the reviewer for the close and thoughtful reading. Your comment helps us clarify the structure and intent of our method more precisely.
>
> Contrary to the concern, our method does not assume sufficiency is fulfilled. Instead, it incorporates a mechanism that **explicitly optimizes sufficiency (PS)** through iterative sampling and evaluation. Specifically, when the original Chain-of-Thought (CoT) is insufficient—i.e., it fails to produce the correct answer—our method **repeatedly executes Algorithm 1 under the same question context to sample new CoTs and re-evaluate their PS**. This iterative resampling increases the chance of obtaining a causally sufficient CoT before applying any necessity-based pruning.
>
> Therefore, **sufficiency is not assumed**, but actively optimized through multiple generations and evaluations. We will revise the manuscript to clarify this mechanism and ensure the abstract and contributions section more accurately reflect this two-phase design.

---

### Official Review · Reviewer_rD2Z · 2025-07-03

**Clarity:** 3
**Significance:** 3
**Originality:** 2
**Rating:** 4
**Confidence:** 3

**Summary:**

This paper proposes a causal framework to enhance CoT reasoning from the perspective of sufficiency and necessity.

Existing COT reasoning cannot ensure that all intermediate steps are useful for the final reasoning, nor can it ensure that the intermediate steps are sufficient to make the final reasoning.

This paper proposes the concept of PNS, which aims to remove redundant steps on the one hand and complete the missing steps on the other hand, so as to improve the performance of reasoning and reduce the cost.

**Questions:**

1. Does the complete PNS consume more tokens than COT? If so, is there any way to directly generate shorter and better reasoning?
2. Analysis of the number and complexity of tokens for complete PNS reasoning
3. Qualitative analysis of whether the final reasoning is sufficient and necessary

**Ethical Concerns:**

["NO or VERY MINOR ethics concerns only"]

**Final Justification:**

Thank you for your additional analysis, we would like to raise our rating to 4

**Limitations:**

yes

**Quality:**

2

**Strengths And Weaknesses:**

**Strengths**
This paper proposes to optimize the reasoning process from the perspective of sufficiency and necessity.
Dealing with the redundancy problem of COT reasoning is indeed a huge challenge, and it is intuitively reasonable to generate sufficient and necessary reasoning processes. Experiments have shown that the PNS method proposed in this paper does improve the reasoning accuracy in the end. The resulting final reasoning length is also shorter than COT reasoning.

**Weaknesses**
1. This paper claims that it significantly reduces reasoning redundancy, but PNS does not seem to be able to directly generate short reasoning results. It checks the entire chain and each step after COT reasoning, rewrites some steps, and finally prune them. Doesn't this process consume more tokens?
2. The framework of this article seems to be prompt engineering. Is there a way to make the model eventually learn to generate short and effective reasoning in one generation?
3. If the PNS reasoning process described in 1 is correct, you may need to add more complexity analysis and cost experiments on the number of all tokens generated during the entire reasoning process.
4. You may need human evaluation or qualitative experiments to analyze whether the final reasoning generated by PNS is sufficient and necessary.

---

> ### Author Rebuttal · Authors · 2025-07-30
>
> **1.[Q1 & W1, W2] Does the complete PNS procedure consume more tokens than CoT? If so, can we reduce the token usage or train the model to generate shorter and more effective reasoning directly? Is this approach just prompt engineering?**
>
> Thank you for the thoughtful questions. We respond as follows:
>
> Our method adopts a **two-stage framework** (see Lines 120–128):
>
> 1. **Data preparation stage**: We apply the PNS-based filtering method to existing CoT data to optimize and clean reasoning traces. This ensures that the selected CoTs satisfy both **necessity and sufficiency**.
> 2. **Inference stage**: We use the optimized dataset to perform **in-context learning** or **fine-tuning**, enabling the model to **learn to generate short and effective reasoning in a single forward pass**.
>
> While it is true that the PNS filtering process incurs additional token usage during the data preparation phase, this is a **one-time cost**. Crucially, **the actual inference process after fine-tuning consumes fewer tokens**, as the model learns to generate concise and accurate reasoning directly—without requiring step-by-step rollout or post-hoc pruning.
>
> In this sense, the strength of our method does **not** lie in reducing token usage during the filtering phase, but rather in reshaping the training distribution. By training on high-PNS data, the model internalizes causally meaningful and non-redundant reasoning patterns. The benefit manifests at inference time, where the model naturally produces streamlined CoTs, improving both efficiency and interpretability.
>
> Finally, we would like to clarify that our method is not merely prompt engineering. Although the filtering procedure is prompt-based, the core of our approach lies in learning from causally sound CoTs via supervised fine-tuning (SFT). After this training, the model is able to generalize the PNS principle, generating short and effective reasoning from scratch—without requiring further intervention.
>
> We will revise the manuscript to clarify this two-stage design and the distinction between one-time filtering overhead and long-term inference efficiency.
>
> **2.[Q2 & W3] Analysis of the Number and Complexity of Tokens for Complete PNS Reasoning.**
>
> (1) We **added a new experiment** (see **Table R1**) showing token cost and rollout complexity across LLMs (e.g., Qwen2.5) and LRMs (e.g., DeepSeek R1) during PNS filtering.
>
> (2) We also compare average tokens per CoT before and after PNS-based fine-tuning (see **Table 3 in the main paper)**, demonstrating reduced inference cost and improved conciseness.
>
> - Across three different LLM families, we observe a consistent trend: while PNS filtering introduces a moderate one-time token cost during data preparation, it yields over 20% token savings during inference.
> - The accuracy of all models is either maintained or improved, confirming that the CoTs retained after causal filtering are both concise and effective.
> - This supports our central claim: PNS filtering reshapes the training distribution, enabling the model to generate more efficient and logically minimal reasoning paths—without requiring inference-time pruning.
>
> **Table R1**. Token usage and complexity during PNS filtering
> | **Model (Base)**            | **Dataset**       | **Rollouts** | **Avg. Tokens** | **Time Complexity** |
> |----------------------------|-------------------|--------------|------------------|---------------------|
> | Qwen-2.5-72B-Instruct | GSM8k             | 9.77         | 1,104.42         | $O(k · n²)$          |
> |                            | CommonsenseQA     | 24.84        | 2,945.61         | $O(k · n²)$           |
> |                            | MATH-500          | 21.66        | 2,315.26         | $O(k · n²)$          |
> |                            | AIME              | 61.81        | 12,145.87        | $O(k · n²)$         |
> | DeepSeek R1            | GSM8k             | 22.62        | 2,861.12         | $O(k · n²)$         |
> |                            | CommonsenseQA     | 30.81        | 4,501.99         | $O(k · n²)$         |
> |                            | MATH-500          | 46.16        | 9,855.70         | $O(k · n²)$         |
> |                            | AIME              | 660.90       | 371,430.57       | $O(k · n²)$          |
> > *Rollouts*: Average number of step-level interventions (regenerations) per CoT.
> *Avg. Tokens*: Average total number of tokens generated per example during PNS filtering.
>
> **Time Complexity Derivation:**
>
> Let $n$ be the number of reasoning steps in a CoT;
>
> $k$ the number of rollouts per step;
>
> $l_{\text{step}}$ the average number of tokens generated per step;
>
> $l_{\text{out}}$ the number of tokens required to generate an answer during verification (typically constant);
>
> and $t_{\text{token}}$ the time required to generate one token (a constant).
>
> Assuming each rollout at step $i$ generates approximately $(n - i)$ subsequent steps, each with $l_{\text{step}}$ tokens, the total rollout cost is:
>
> $$
> T_{\text{rollout}} = \sum_{i=1}^{n} k \cdot (n - i) \cdot l_{\text{step}} \cdot t_{\text{token}} = O(k \cdot l_{\text{step}} \cdot t_{\text{token}} \cdot n^2)
> $$
>
> Each rollout also requires evaluating whether the generated result is valid, including generating the answer ($l_{\text{out}}$ tokens) and the full chain:
>
> $$
> T_{\text{eval}} = \sum_{i=1}^{n} k \cdot \left[ (n - i) \cdot l_{\text{step}} + l_{\text{out}} \right] \cdot t_{\text{token}} = O(k \cdot t_{\text{token}} \cdot (l_{\text{step}} \cdot n^2 + l_{\text{out}} \cdot n))
> $$
>
> Since $l_{\text{out}} \ll l_{\text{step}} \cdot n$, we simplify:
>
> $$
> T_{\text{eval}} = O(k \cdot l_{\text{step}} \cdot t_{\text{token}} \cdot n^2)
> $$
>
> Combining both rollout and evaluation:
>
> $$
> T_{\text{total}} = T_{\text{rollout}} + T_{\text{eval}} = O(k \cdot l_{\text{step}} \cdot t_{\text{token}} \cdot n^2)
> $$
>
> Since $l_{\text{step}}$ and $t_{\text{token}}$ are typically constant or nearly so, the final complexity simplifies to:
>
> $$
> T_{\text{PNS}} = O(k \cdot n^2)
> $$
>
> Overall, our results show that the proposed method **reduces reasoning redundancy**, trading a one-time overhead for sustained improvements in inference efficiency.
>
> **3.[Q3 & W4] Qualitative Analysis of Whether the Final Reasoning Is Sufficient and Necessary.**
>
> Thank you for the insightful suggestion. We conducted **an additional human evaluation** on 50 CoT samples, all generated by a model fine-tuned on data optimized using our PNS-based algorithm. Five mathematics experts were involved in validating the 50 samples, assessing whether each reasoning chain was sufficient and necessary to support the final answer.
>
> - **S&N**: The reasoning is both sufficient and necessary to support the final answer;
> - **SbU**: The reasoning is sufficient but includes unnecessary (redundant) steps;
> - **NbI**: The reasoning is insufficient, i.e., missing critical steps;
>
> The results are summarized as follows:
>
> **Table R2.** Human Evaluation of Reasoning Quality
>
> | **Dataset**      | **# Samples** | **Fully Sufficient** | **Redundant** | **S&N**        | **SbU**       | **Nbl**       |
> |------------------|---------------|-----------------------|----------------|----------------|----------------|----------------|
> | GSM–8k           | 20            | 19                    | 3              | 17             | 2              | 1              |
> | Commonsense QA   | 15            | 15                    | 1              | 14             | 1              | 0              |
> | MATH–500         | 15            | 13                    | 4              | 11             | 2              | 2              |
> | **Total**        | **50**        | **47 (94.0%)**        | **8 (16.0%)**  | **42 (84.0%)** | **5 (10.0%)**  | **3 (6.0%)**   |
>
> These results confirm that most model outputs are not only accurate but also logically sound under the criteria of sufficiency and necessity.
>
> Notably:
>
> - 84% of CoTs are classified as both sufficient and necessary (S&N),
> - Only 3 out of 50 CoTs (6.0%) were deemed insufficient (Nbl),
> - MATH-500 exhibits a slightly higher rate of redundancy and incompleteness, which is expected due to the complexity of mathematical reasoning.
>
> We will include this evaluation and Table R2 in the revised manuscript.

---

> > ### Author Response · Authors · 2025-08-05
> >
> > Dear Reviewer rD2Z,
> >
> > As the discussion deadline approaches, we are wondering whether our responses have properly addressed your concerns? Your feedback would be extremely helpful to us. If you have further comments or questions, we hope for the opportunity to respond to them.
> >
> > Many thanks,
> >
> > 2837 Authors

---

> > > ### Comment · Reviewer_rD2Z · 2025-08-07
> > >
> > > Thank you for your additional analysis, we would like to raise our rating to 4

---

> ### Author Response · Authors · 2025-08-04
> **We would like to supplement more experiment results to fully address your concerns on whether our model indeed learns to generate shorter and more effective reasoning traces**
>
> We would like to **supplement more experiment results** to fully address your concerns on whether our model indeed learns to generate shorter and more effective reasoning traces—beyond prompt engineering or manual filtering.
>
> We conducted a semantic redundancy analysis on three model families: **DeepSeek-R1-Distill-Qwen-1.5B, Phi-4-mini-reasoning, and DeepScaleR-1.5B-Preview**. Each model was evaluated before and after PNS fine-tuning using 15 GSM8k and CommonsenseQA questions. We measured average steps per CoT, redundancy score (via sentence embedding similarity), and expert-rated logical validity.
>
> **Table R3.** Semantic Redundancy Analysis Across Models
>
> | **Model**                       | **Avg. Steps ↓** | **Redundancy Score ↓** | **Valid Reasoning (%) ↑** |
> | ------------------------------- | ---------------- | ---------------------- | ------------------------- |
> | DeepSeek-R1 (Base)              | 16.8             | 0.81                   | 80.0%                     |
> | DeepSeek-R1 (PNS-Finetuned)     | 12.4             | 0.42                   | 86.7%                     |
> | Phi-4-mini (Base)               | 21.2             | 0.74                   | 73.3%                     |
> | Phi-4-mini (PNS-Finetuned)      | 13.6             | 0.39                   | 93.3%                     |
> | DeepScaleR-1.5B (Base)          | 14.5             | 0.77                   | 66.7%                     |
> | DeepScaleR-1.5B (PNS-Finetuned) | 10.1             | 0.44                   | 80.0%                     |
>
> Results show that across all three models, **PNS-finetuned versions consistently generate shorter and less redundant reasoning while improving logical validity**. This demonstrates that the model has learned to eliminate trivial steps and focus on causally essential information.

---

> ### Author Response · Authors · 2025-08-04
> **We added a supplementary analysis to better address the issue of token usage and reasoning complexity in complete PNS filtering.**
>
> We **added a supplementary analysis** to better address the issue of token usage and reasoning complexity in complete PNS filtering. Specifically, we introduced two new metrics—**Original Step Length and Original Token Count**—to more precisely capture the input CoT complexity before filtering. This extended analysis, shown in Table R4, offers deeper insight into how input length correlates with PNS overhead.
>
> **Table R4**. Token usage and complexity during PNS filtering
>
> | **Model (Base)**      | **Dataset**   | **Original Step Length** | **Original Tokens** | **Rollouts** | **Avg. Tokens** | **Time Complexity** |
> | :-------------------- | :------------ | :----------------------- | :----------------------- | :----------- | :-------------- | :------------------ |
> | Qwen-2.5-72B-Instruct | GSM8k         | 8.1                      | 113.8                    | 9.77         | 1,104.42        | $O(k \cdot n^2)$    |
> |                       | CommonsenseQA | 3.7                      | 109.2                    | 24.84        | 2,945.61        | $O(k \cdot n^2)$    |
> |                       | MATH-500      | 9.2                      | 281.8                    | 21.66        | 2,315.26        | $O(k \cdot n^2)$    |
> |                       | AIME          | 12.5                     | 531.3                    | 61.81        | 12,145.87       | $O(k \cdot n^2)$    |
> | DeepSeek R1           | GSM8k         | 5.4                      | 137.3                    | 22.62        | 2,861.12        | $O(k \cdot n^2)$    |
> |                       | CommonsenseQA | 6.2                      | 191.0                    | 30.81        | 4,501.99        | $O(k \cdot n^2)$    |
> |                       | MATH-500      | 16.2                     | 387.6                    | 46.16        | 9,855.70        | $O(k \cdot n^2)$    |
> |                       | AIME          | 120.3                    | 2438.8                   | 660.90       | 371,430.57      | $O(k \cdot n^2)$    |
>
> Contrary to intuition, our findings show that **longer original CoTs do not necessarily incur higher PNS cost.** Token usage during filtering varies with the causal structure, not just surface length, demonstrating that PNS adapts dynamically rather than scaling linearly with input size.
>
> As before, **this cost is incurred only once during the offline data cleaning phase**. After fine-tuning, the model learns to produce short, causally sufficient CoTs without any inference-time rollouts—delivering both efficiency and effectiveness.

---

### Official Review · Reviewer_Uvzq · 2025-07-07

**Clarity:** 2
**Significance:** 3
**Originality:** 2
**Rating:** 4
**Confidence:** 3

**Summary:**

This paper introduces a causal framework for optimizing Chain-of-Thought (CoT) reasoning by identifying and preserving only causally sufficient and necessary steps. Using counterfactual metrics (PS and PN), the method prunes redundant reasoning from correct CoTs and applies the optimized traces for in-context learning and fine-tuning. Experiments show improved reasoning efficiency and accuracy across multiple benchmarks.

**Questions:**

Please refer to the “weakness”.

**Ethical Concerns:**

["NO or VERY MINOR ethics concerns only"]

**Final Justification:**

The rebuttals have solved most of my concerns, I decide to keep my positive score.

**Limitations:**

yes

**Quality:**

2

**Strengths And Weaknesses:**

Strengths:

The paper presents a clear problem setup and task formulation; the proposed method is straightforward and interpretable; the experimental results appear promising.


Weaknesses:

1.	Limited Novelty.
Using counterfactual interventions like PS and PN in CoT is conceptually interesting, but causal counterfactual as a core mechanism has been adopted in many existing studies across causal inference and CoT interpretability.

2. Algorithm Soundness.
a. Section 4 describes the proposed method as a two-stage optimization process: first enhancing PS, then improving PN. However, the paper does not clearly explain how PS is actually enhanced. Algorithm 1 does not seem to contain any explicit procedure for stage 1’s improving PS, which is confusing.
b. It is also unclear how the method handles initially incorrect CoTs. According to Algorithm 1, if the original CoT is not sufficient (i.e., leads to an incorrect answer), the algorithm simply returns the unmodified chain. This seems to limit the practical applicability of the method. If the method is indeed capable of improving incorrect CoTs in practice, the authors should provide a clear explanation or clarification in the main text.

3. On experimental setup and conclusions.
a. A weakness of the experimental part may be the lack of detail regarding metric definitions and inference setup, e.g., it is unclear how accuracy is calculated—was it based on a single rollout, or averaged over multiple rollouts? Additionally, the decoding setup is a bit confusing: the appendix states that inference was performed with temperature=0.6, top_p=0.95, and greedy decoding, which appear to be contradictory.
b. Regarding RQ1, I am confused by the conclusion that “the PNS-based algorithm can improve accuracy.” According to Algorithm 1, the proposed method appears to apply only to CoTs that already lead to correct answers (i.e., PS = 1). This raises a concern: if only correct CoTs are subject to optimized, then the upper bound should be the original accuracy. I suggest that the authors further clarify the metric calculation.
c. Regarding RQ2, the authors mention that all SFT examples were manually verified. What is the reasoning behind this, and could it potentially impact the comparison fairness? In other words, if human filtering is still required, it undermines the method’s automation and scalability. Additionally, the paper does not clarify whether the Noncausal SFT baseline was trained only on correct CoTs (or with wrong CoTs included), which significantly affects the performance of SFT and the fairness of the comparison.

---

> ### Author Rebuttal · Authors · 2025-07-30
>
> **1. [W1] Related Work Discussion about Using PS and PN in CoT.**
>
> Thank you for your insightful comment. To the best of our knowledge, this work is the first to formally introduce the causal concept of Probability of Necessity and Sufficiency (PNS) into large language models, specifically to address the issue of “overthinking” in Chain-of-Thought (CoT) reasoning. While prior studies have explored counterfactual metrics such as PS and PN, PNS has not been applied in this context, nor has it been employed as a causal criterion to identify and prune unnecessary reasoning steps. Our approach offers a novel and actionable causal perspective, leveraging effective intervention strategies to improve both the performance and interpretability of the model.
>
> **2.[W2] In Algorithm 1, how is PS improved if the original CoT is incorrect?**
>
> Thank you for the question. As PS is defined over the entire reasoning chain (CoT), while PN is defined at the step level, our approach **aims to improve PS by repeatedly executing Algorithm 1 under the same question context**, especially when the original CoT is insufficient (i.e., fails to yield the correct answer).
>
> In each execution, the model samples an alternative CoT, and its PS is re-evaluated. This repeated sampling increases the likelihood of obtaining a CoT with higher PS, thereby enhancing the reliability of the subsequent PN-based intervention.
>
> Although Algorithm 1 does not explicitly separate this process as a standalone stage, it plays a key role in improving PS before any pruning is applied. We will clarify this iterative refinement mechanism in the revised manuscript to better reflect the actual implementation and practical utility of our method.
>
> **3.[W3] Details about metric definition and inference setup (e.g., accuracy calculation and decoding setup)**
>
> Thank you for your question. We address each point as follows:
>
> **a. On metric definition and decoding setup:**
>
> The reported **accuracy** refers to the average proportion of correctly answered instances in the test set. It is computed as: $\text{Accuracy} = \frac{\text{Number of correct CoTs}}{\text{Total number of CoTs in the dataset}}$
>
> This is a standard evaluation setup in LLM reasoning benchmarks and is consistent with prior CoT literature.
>
> Regarding the decoding configuration, the mention of greedy decoding in the appendix was a typo. We apologize for the confusion. In practice, we consistently use temperature = 0.6 and top_p = 0.95 across all experiments.
>
> **b. On the meaning of “accuracy improvement” in RQ1:**
>
> The accuracy gains reported in RQ1 result from the fact that, in practice, we continuously apply Algorithm 1 across the dataset. Through repeated rollouts of each CoT step, the probability of generating a correct answer increases significantly. A CoT with PS < 1 might be overlooked in a single run of Algorithm 1, but is revisited and re-evaluated in subsequent rollouts until the entire dataset undergoes PNS-based optimization.
>
> **c. On fairness concerns regarding manual inspection in RQ2:**
>
> We confirm that **no manual editing was performed on the CoTs selected by our method**—only human verification of quality was conducted. As shown in RQ1, the CoTs filtered via the PNS method are already causally sound and can be directly used for SFT.
>
> The **Non-Causal SFT baseline** uses raw, unfiltered CoTs (including both correct and incorrect samples). In contrast, the **Causal SFT baseline** uses CoTs that were optimized via the PNS-based filtering procedure. This distinction is **intentional and fair**, as our goal is to demonstrate how causal filtering improves data quality and, ultimately, downstream performance.
>
> We will clarify this design choice and its rationale in greater detail in the revised manuscript.

---

> > ### Author Response · Authors · 2025-08-05
> >
> > Dear Reviewer Uvzq,
> >
> > As the discussion deadline approaches, we are wondering whether our responses have properly addressed your concerns? Your feedback would be extremely helpful to us. If you have further comments or questions, we hope for the opportunity to respond to them.
> >
> > Many thanks,
> >
> > 2837 Authors

---

> > ### Comment · Reviewer_Uvzq · 2025-08-06
> >
> > Thanks for the thorough reply, my concerns are mostly solved. I decide to keep my positive score.

---

### Note · Authors · 2025-08-14

Dear AC and reviewers,

We sincerely appreciate the considerable time and effort you have dedicated in evaluating our work. After reviewer-author discussion, we are encouraged that **all four reviewers are on the positive side of acceptance.** To facilitate the upcoming ac-reviewer discussion, we summarize the up-to-date status as follows.

- **Reviewer Uvzq** acknowledged that ```the thorough reply has resolved my concerns``` and he/she ```decide to keep my positive score```.  During rebuttal, we addressed the “*experimental setup and conclusions*” concern by clarifying metric definitions and decoding, and highlighted that no manual editing was performed on the CoTs, ensuring the fairness of the comparison.  For addressing "*how is PS improved*", we clarified PS improves by iteratively resampling and re-evaluating CoTs, with the experimental results shown in Table 1.
- **Reviewer rD2Z** praised that ```we would like to raise our rating to 4```.  For addressing concerns on “*Does the complete PNS consume more tokens than CoT*”, we added complexity and cost analyses showing our two-stage framework incurs a one-time cost during filtering, but achieving > 20% inference token savings. We also added new experiments on Qwen2.5 and DeepSeek R1 for supplementing "*analysis of the number and complexity of tokens for complete PNS reasoning*"— we will include these in our final version.
- **Reviewer Br1J** replied that ```my primary concerns have been addressed``` and he/she  ```also read the other reviews and the ensuing discussions, which I found quite helpful```, and ```will revise my score upward to reflect this.```  In response to “*how our method implements causal reasoning*”, we provided a mathematical explanation of the causal reasoning part of our method, and added experiments to address the “*discussion of the complexity*” concern.
- **Reviewer iGwQ** commented that ```the proper response has addressed my concern on lack of baselines``` and further asked: *“Can the authors explain where sufficiency is optimized in their proposed method?”*.  In response to this, we clarified that sufficiency (PS) improves by iteratively resampling and re-evaluating CoTs, and we explained the detailed procedure —  thank you so much!

Please review our detailed responses to each point raised. We are confident that our responses can thoroughly address the reviewers' concerns. Thank you so much for the great efforts and thoughtfulness — we truly appreciate it!

Authors of #2837

---

### Decision · Program_Chairs · 2025-09-17

**Decision:**

Accept (poster)

**Comment:**

This paper presents a causal framework for optimizing CoT reasoning by evaluating step-wise sufficiency and necessity using PNS. Through a two-stage process—resampling to improve sufficiency and pruning via necessity—the method improves both reasoning accuracy and efficiency. Reviewers raised concerns about novelty, clarity, and evaluation, but the authors responded with detailed explanations, new experiments, and additional baselines. Reviewers acknowledged the thorough rebuttal and some of them raised their scores. Therefore I recommend acceptance.